# STREAMAGENT: TOWARDS ANTICIPATORY AGENTS FOR STREAMING VIDEO UNDERSTANDING

## ABSTRACT

Real-time streaming video understanding in domains such as autonomous driving and intelligent surveillance poses challenges beyond conventional offline video processing, requiring continuous perception, proactive decision making, and responsive interaction based on dynamically evolving visual content. However, existing methods rely on alternating perception-reaction or asynchronous triggers, lacking *task-driven planning* and *future anticipation*, which limits their real-time responsiveness and proactive decision making in evolving video streams. To this end, we propose a **StreamAgent** that anticipates the temporal intervals and spatial regions expected to contain future task-relevant information to enable proactive and goal-driven responses. Specifically, we integrate question semantics and historical observations through prompting the anticipatory agent to anticipate the temporal progression of key events, align current observations with the expected future evidence, and subsequently adjust the perception action (*e.g.,* attending to task-relevant regions or continuously tracking in subsequent frames). To enable efficient inference, we design a *streaming KV-cache memory* mechanism that constructs a hierarchical memory structure for selective recall of relevant tokens, enabling efficient semantic retrieval while reducing the overhead of storing all tokens in the traditional KV-cache. Extensive experiments on streaming and long video understanding tasks demonstrate that our method outperforms existing methods in response accuracy and real-time efficiency, highlighting its practical value for real-world streaming scenarios. Code is available in the supplementary material.

## 1 INTRODUCTION

As vision models are increasingly deployed in real-world scenarios (*e.g.,* autonomous driving and intelligent surveillance), research has progressively shifted towards understanding continuous video streams. Streaming video understanding requires continuous processing of incoming frames, efficient extraction of critical information, proactive decision-making, and responsive interaction with dynamically evolving visual content, guided by explicitly defined queries.

Recent works, such as VideoStreaming (Qian et al., 2024), Flash-VStream (Zhang et al., 2024a), and ReKV (Di et al., 2025) have explored multimodal large language models (MLLMs) (Touvron et al., 2023; OpenAI, 2023c) for streaming video understanding, leveraging memory-based mechanisms and aligned instruction-tuning strategies to process long video streams and support real-time scene. However, beyond continuous perception, these models require a proactive response mechanism to determine whether to respond immediately or continue observing to gather sufficient evidence. Therefore, *we argue that effective proactive response in streaming scenarios relies on backward tracking of historical context, real-time understanding and reasoning over current content, and forward active responding through timely deferral until sufficient future information becomes available.*

To enable proactive interaction, recent studies such as VideoLLM-online (Chen et al., 2024a) adopt an alternating "perception–reaction" mechanism, as illustrated in Figtu re. 1 (a), where a single LLM is responsible for both video perception and reaction. However, constrained by the autoregressive nature of LLMs, perception and generation cannot be performed in parallel, which delays subsequent frame processing and reduces system responsiveness. Therefore, Dispider (Qian et al., 2025) attempts to improve response efficiency by decoupling decision and reaction, employing a lightweight asynchronous binary trigger. However, these methods lack task-driven proactive planning and anticipation of the future, which limits their real-time responsiveness and proactive decision-making in evolving

Figure 1: **Comparison between StreamAgent and existing methods.** Prior online methods (Chen et al., 2024a; Qian et al., 2025) enable proactive interactions but rely on either (a) alternating interaction, causing slow processing, or (b) asynchronous binary triggers, leading to inaccurate responses. In contrast, **StreamAgent** integrates continuous perception with task-driven planning and future anticipation, enabling proactive identification of key temporal and spatial cues, and supporting efficient asynchronous reaction through the **streaming KV-cache**.

video streams. As illustrated in Figure 1 (b), Dispider prematurely predicts (*i.e.,* "wall"), a mistake rooted in limited temporal reasoning and inadequate modeling of past and anticipated future signals, leading to the failure to identify the actual referent (*i.e.,* "painting").

In this work, we propose a **StreamAgent** that anticipates the temporal intervals and spatial regions expected to contain future task-relevant information to enable proactive and goal-driven responses, as illustrated in Figure 1 (c). Specifically, we integrate question semantics and historical observations through prompting the lightweight anticipatory agent to anticipate the temporal progression and spatial locations of key events. Subsequently, the current observation status is aligned with the anticipated progression to determine whether sufficient information has been accumulated to trigger a response. If not, StreamAgent proactively refines its perception strategy based on the predicted trajectory (*e.g.,* attending to task-relevant regions or continuously tracking subsequent frames). Consequently, as new video streams arrive, StreamAgent iteratively updates its spatiotemporal focus to gradually accumulate sufficient evidence for accurate responses generation, thereby enhancing robustness and responsiveness in streaming scenarios. Furthermore, an incremental memory update mechanism, specifically tailored for dynamic streaming inputs, enables the continuous integration of new information, ensuring timely and coherent responses throughout the entire video stream.

To address the long-context bottleneck in streaming (Di et al., 2025) while ensuring efficient inference, we propose a **streaming KV-cache** that leverages the inherent temporal nature of video streams to construct a hierarchical memory structure for adaptive context retrieval, thereby effectively balancing retention and efficiency. Specifically, each video clip is encoded into key-value cache via chunk-wise incremental prefill, enabling retrieval based on query relevance. To enable efficient memory retrieval, our memory mechanism adopts two key designs: *(i)* it maintains short-term memory on the GPU to track ongoing events. *(ii)* it leverages CPU memory to store long-term KV-caches, enabling fine-grained frame-level relevance identification and layer-adaptive retrieval, while alleviating GPU memory constraints in streaming video understanding. Furthermore, the retrieval process dynamically adjusts the number of KV-cache entries per layer based on attention patterns, which reflect how broadly or narrowly each layer attends to past information across the temporal dimension of the video stream. By scoring relevance at the video-frame level and discarding low-importance tokens, this mechanism ensures that only semantically pertinent content is recalled, thereby ultimately enabling accurate and efficient reasoning over long temporal horizons.

We further validate the generalization capability of our method through zero-shot video question answering on streaming and offline video understanding benchmarks, thereby demonstrating robustness across diverse scenarios. In summary, our key contributions are as follows:

- We propose a **StreamAgent** for streaming video understanding that anticipates future task-relevant temporal intervals and spatial regions to enable proactive responses.
- We propose a **streaming KV-cache** that constructs a hierarchical memory structure to enable efficient semantic retrieval and *selective recall* of relevant tokens.
- Experiments demonstrate that our method achieves state-of-the-art performance and yields significant improvements on streaming and long video understanding benchmarks.

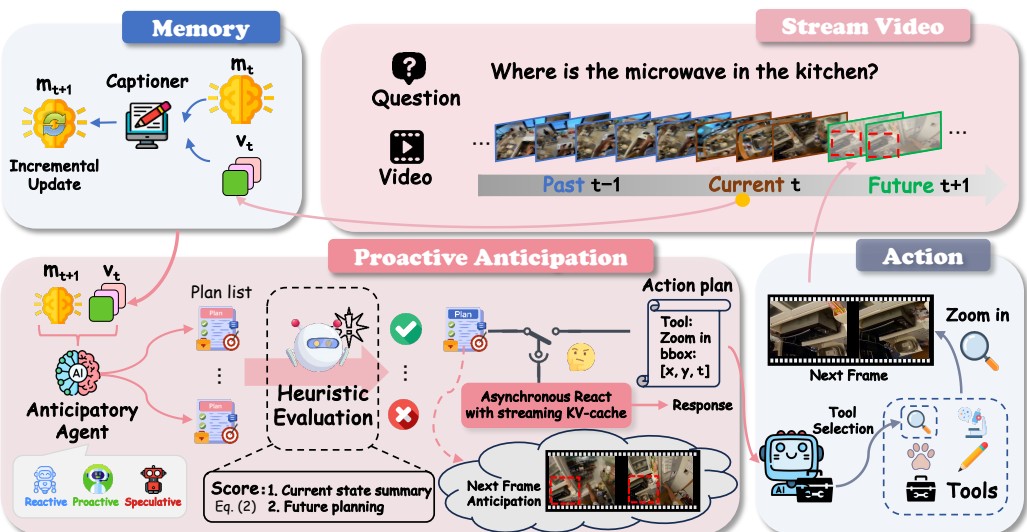

Figure 2: **Overview of the StreamAgent framework:** At each timestep, the system incrementally updates memory based on the current streaming video clip $v_t$ and memory $m_t$ via captioning in a Markov process. Conditioned on the updated memory and current observation, the planner generates multiple future-aware plans through proactive anticipation simulating incoming videos, spanning three perspectives, *Reactive, Proactive, and Speculative*. These plans are scored via heuristic evaluation (Eq. 2), jointly considering current situational awareness and predicted future utility. The selected plan either triggers asynchronous reactive behavior or initiates proactive information hunting by invoking tools (*e.g.,* zooming in task-relevant regions in subsequent frames), enabling goal-directed and proactive responses.

## 2 METHODOLOGY

### 2.1 OVERVIEW OF STREAMAGENT FRAMEWORK

To tackle the challenges of Streaming VideoQA, we propose StreamAgent, a decision agent that anticipates future temporal intervals and spatial regions to contain task-relevant information, enabling proactive responses to user queries, as illustrated in Figure 2.

**Problem Fomulation.** Streaming VideoQA (Ding et al., 2025) is a task designed for answering proactive (Ning et al., 2024), context-sensitive, and real-time questions over video streams. Given a sequence of video stream up to time $T$ be $\mathcal{V}^T = \{v_t\}_{t=1}^T$, where $v_t$ represents the $t$-th video clip aligned with timestamp $t$. Let $Q_t$ denotes a question at a specific time stamp $t$. The goal of the streaming VideoQA is to take advantage of the current content $v_t$ and the past video $\mathcal{V}_{1:t-1}$ to answer the $Q_t$. When video content $\mathcal{V}_{1:t}$ is insufficient to answer the user's question $Q_t$, the model should be able to respond proactively at a future time stamp $t'(t \leq t' \leq T)$ when updated video content $\mathcal{V}_{t+1:t'}$ is sufficient to answer $Q_t$. We define a decision function $D$ to determine whether to wait or respond and a response function $f$ to generate the response. At each timestamp $t$, the model employs $D(Q_t, \mathcal{V}_{1:t})$ to decide and invokes the response function $f$ only when the information is sufficient.

**Incremental Memory Update.** To ensure timely and coherent responses in streaming scenarios, we propose an incremental memory update mechanism to process continuous video streams, as shown in Figure 2. StreamAgent maintains a single memory state $m_t$, which is iteratively updated to summarize the observed data over time. To ensure the efficiency of the decision model, we convert long video token sequences $\mathcal{X}_t = \{x_i\}_{i=1}^{N_t}$ into textual caption tokens $\mathcal{C}_t = \{c_i\}_{i=1}^{n_t}$ as memory. $N_t$ and $n_t$ represent the counts of video and memory tokens at time $t$, where $N_t \gg n_t$. The memory $\mathcal{C}_t$ is updated continuously as the video stream evolves. Assuming the Markov property, the memory state $m_t$ at time $t$ depends solely on the immediate previous memory state $m_{t-1}$ and the current video input $v_t$ (Yang et al., 2025b), rather than the full history, a process formalized as:

$$p(m_t|m_{t-1}, v_t) = p(m_t|m_{\tau<t}, v_{\tau \leq t}), \qquad (1)$$

where $p(\cdot)$ represents the conditional probability distribution over memory states. By continuously updating the memory with each incoming video clip $v_t$, the agent ensures that the memory state remains consistent and coherent throughout the entire video stream.

**Proactive Anticipation with Heuristic Evaluation.** Within the StreamAgent framework, we introduce the anticipatory stream agent, which leverages a lightweight Multi-modal Language Model, denoted as $A(\cdot)$, to forecast the temporal dynamics and spatial distribution of task-relevant events. The current state is represented as $S_t = A(m_t, v_t)$, based on memory $m_t$ (text) and $v_t$ (video). From $S_t$, we estimate the transition distribution $p(E_t \mid S_t)$, which predicts future event sequences $E_t = \{e_i\}_{i=t+1}^{t'}$. To simulate diverse future anticipation capabilities, StreamAgent adopts a *multi-perspective planning mechanism* that prompts the agent to plan from three complementary modes: *Reactive*, *Proactive*, and *Speculative*. The *Reactive* mode grounds decisions in currently observed evidence, emphasizing certainty-prioritized decisions based on established facts. The *Proactive* mode extrapolates from current observations to anticipate near-future outcomes, actively predicting potential future results for faster responses, while the *Speculative* mode ventures beyond available evidence to explore long-term possibilities under high uncertainty. Formally, these planning modes collectively yield a set of $k$ candidate plans:

$$\mathcal{P} = \{P_j\}_{j=1}^{k}, \qquad P_j = \Pi(S_t, E_t \mid m_t, v_t, \text{mode} = j), \tag{2}$$

where $\Pi(\cdot)$ denotes the planning function, and each $P_j$ corresponds to a trajectory simulating potential responses to incoming video streams under the $j$-th planning perspective. To select the optimal plan, StreamAgent employs a heuristic scoring function inspired by the A* algorithm:

$$F(P_j \mid S_t, E_t, m_t, v_t) = G(P_j, S_t, m_t, v_t) + \lambda \cdot U(P_j, E_t, S_t, m_t, v_t), \tag{3}$$

where $G(\cdot)$ evaluates the current (immediate) utility, $U(\cdot)$ estimates the future (anticipated) utility, and $\lambda$ is a weighting factor balancing the two. The optimal plan is then given by

$$\hat{P}(t) = \underset{P_j \in \mathcal{P}(S_t, E_t)}{\arg\max} \; F(P_j \mid S_t, E_t, m_t, v_t). \tag{4}$$

As the new video $v_{t+1}$ arrives, the agent doesn't just make a one-time prediction; it iteratively refines the existing plan. This continuous integration of new visual information allows StreamAgent to maintain a dynamic and adaptive planning capability, ensuring its strategies evolve in real-time to match the ever-changing streaming video environment. Based on the optimal plan $\hat{P}$, the anticipatory agent, decides whether to trigger an asynchronous response to answer the question.

**Tool-Augmented Action.** To achieve true active perception (Fu et al., 2025; Li et al., 2025c; Wang et al., 2024c), StreamAgent is not merely a passive consumer of video frames, but a goal-driven information explorer. Leveraging a suite of external tools (Hu et al., 2024b; Wu et al., 2025b), its tool-augmented reasoning mechanism proactively determines when, where, and how to acquire critical information, as shown in Figure 2. Specifically, given the user query $Q$, the predicted reasoning plan $\hat{P}$ and the current state $S_t$, the action agent proactively selects a subset of tools $\mathcal{T}' \subseteq \mathcal{T} = \{\phi_i\}_{i=1}^{M}$ based on its planning needs, where $M = |\mathcal{T}|$, and apply them to the incoming video clip $v_{t+1}$. At each time stamp $t$, the agent selects a subset of the tool set $\mathcal{T}'_t \subseteq \mathcal{T}$ via a tool selection policy $\pi_{\text{tool}}$, formulated as:

$$\mathcal{T}'_t = \pi_{\text{tool}}(S_t, \hat{P}, Q), \tag{5}$$

For each selected $\theta_i \in \mathcal{T}'_t$, a specific video sub-region or frame window $v_{t+1}^{j} \subseteq v_{t+1}$ is determined by a policy $\pi_{\text{crop}}$:

$$v_{t+1}^{j} = \pi_{\text{crop}}(S_t, \hat{P}, \phi_j), \quad R_j = \phi_j(v_{t+1}^{j}). \tag{6}$$

The intermediate results from all tools form a result set: $\mathcal{R}_t = \{R_j\}_{j=1}^{J}$, where $J = |\mathcal{T}'|$ is the number of tools selected at time $t$. The updated perception state is computed by passing the memory and tool results into the anticipatory agent $A(\cdot)$:

$$S_{t+1} = A(m_t, \mathcal{R}_t). \tag{7}$$

To ensure efficient and targeted responses, StreamAgent invokes the responding model only when the video stream ends (*i.e.,* $t = T$) or when sufficient information has been accumulated to answer the query $Q$. By iteratively planning tool usage $\mathcal{T}'_t$ and refining the perception targets $v_{t+1}$, StreamAgent exhibits proactive information hunting behavior, dynamically adapting its sensory strategy to prioritize data accelerating progress along the predicted planning trajectory $\hat{P}$. This targeted and tool-augmented approach enables efficient planning in long-horizon, streaming video environments.

Figure 3: **Overview of streaming KV-caches. (a) Long-term Memory:** As the video stream is continuously encoded, key-value (KV) caches from earlier frames are offloaded to CPU memory as long-term memory. **(b) Selective Recall:** Upon a query, relevant KV-caches are dynamically selected across layers based on attention scores (within $\alpha$ of the max). **(c) Short-term Memory:** Selected KV-caches are reloaded to GPU and combined with streaming inputs as short-term memory for efficient response generation.

## 2.2 STREAMING VIDEO KV-CACHES MEMORY MECHANISM

This section presents the two processes, *Continuous Perception* and *Selective Recall*, that constitute the streaming KV-caches memory mechanism, as illustrated in Figure 3.

**Continuous Perception.** Continuous Perception adopts an incremental encoding and prefill strategy, storing the resulting KV caches from each video clip into long-term memory for future retrieval (*i.e.,* Selective Recall). As shown in Figure 3 (a), each video clip is sequentially encoded and prefilled, producing a KV-cache: $\{H_{\text{clip}}^k, H_{\text{clip}}^v\} = \{(\mathbf{k}_j, \mathbf{v}_j)\}_{j=1}^{l_P}$, where $l_P$ denotes the total KV length from prior clips. To minimize peak activation memory usage, we apply chunked prefill within each clip. Given an incoming video clip $v_i$ with token sequence $\mathbf{Z}^{v_i} = \{z_j^{v_i}\}_{j=1}^n$, we divide it into $C$ chunks $\mathbf{C}^{v_i} = \{\mathbf{Z}_j^{v_i}\}_{j=1}^\mu$, where $\mu$ and $n$ denote the total number of chunks and tokens in $v_i$, respectively, and each token $z_j^{v_i} \in \mathbb{R}^h$ has hidden size $h$. At each transformer layer, we sequentially prefill the current visual tokens chunk $X = \{z_{j+l_C}\}_{j=1}^{l_X}$ while maintaining the corresponding KV-cache of the past chunks as $\{H_{\text{chunk}}^k, H_{\text{chunk}}^v\} = \{(\mathbf{k}_j, \mathbf{v}_j)\}_{j=1}^{l_C}$, where $l_X$ represents the chunk size, and $l_C$ denotes the length of the previously accumulated KV cache in the current clip. the attention keys and values are then built as:

$$K = [H_{\text{clip}}^k, H_{\text{chunk}}^k, XW_K], V = [H_{\text{clip}}^v, H_{\text{chunk}}^v, XW_V], \tag{8}$$

where $W_K$, and $W_V$ are the $K$ and $V$ weight matrices, stored for future retrieval, respectively.

**Selective Recall.** Selective Recall dynamically retrieves question-relevant KV-caches from long-term memory into GPU-resident short-term memory, in contrast to the recent fixed top-k strategy (Di et al., 2025), which often retains irrelevant or redundant information, thereby leading to suboptimal overall memory utilization. This adaptive mechanism further aligns with the attention behavior of each transformer layer, ultimately enabling more efficient and accurate inference.

• *Retrieving with Dynamic Attention Pattern.* Recent advances in long-sequence modeling with LLMs show that the number of influential keys and values varies across different layers (Lee et al., 2024b). Therefore, unlike ReKV (Di et al., 2025), which retains a fixed number of KV entries per layer, we propose a layer-adaptive retrieval strategy that dynamically selects variable numbers of KV entries based on attention patterns (see Figure 3 (b)). This allows layers with broader attention to access more entries, while layers with narrower focus retrieve fewer. In detail, during video stream prefill, we we first compute a representative feature per frame to enable rapid scoring: $\frac{1}{T_f} \sum_{j=1}^{T_f} \mathbf{k}_j \in \mathbb{R}^{D'}$, where $T_f$ is the number of tokens per frame, and $\mathbf{k}_j$ is the $j$-th key vector. Subsequently, to match frame relevance to the question, we compute the query-level attention descriptor by averaging over all question tokens: $\frac{1}{T_q} \sum_{j=1}^{T_q} \mathbf{q}_j \in \mathbb{R}^{D'}$, where $T_q$ is the number of query tokens, and $\mathbf{q}_j$ is the score vector of the $j$-th token across each attention head. Since softmax exponentially amplifies higher attention scores, we filter out less relevant frames by retaining only those with scores within within a margin $\alpha$ of the maximum, *i.e.,* score $\geq \max -\alpha$. The set of important keys can be formulated as:

$$\mathcal{J}_h = \left\{ j \in \{1, \ldots, L_k\} \mid \max_{j'}(\mathbf{S}_h) - \mathbf{S}_{h,j} \leq \alpha \right\}, \tag{9}$$

where $\mathbf{S}_h \in \mathbb{R}^{L_k}$ and $L_k$ denotes the attention scores for head $h$ and total number of stored keys, respectively. For example, consider $l$ frames, with $n$ frames having attention scores above the

| Model | #Frames | Real-Time Visual Perception | | | | | | | Backward Tracing | | | | Forward Active Responding | | | | Overall |
|---|---|---|---|---|---|---|---|---|---|---|---|---|---|---|---|---|---|
| | | OCR | ACR | ATR | STU | FPD | OJR | Avg. | EPM | ASI | HLD | Avg. | REC | SSR | CRR | Avg. | Avg. |
| Human Agents | - | 94.0 | 92.6 | 94.8 | 92.7 | 91.1 | 94.0 | 93.2 | 92.6 | 93.0 | 91.4 | 92.3 | 95.5 | 89.7 | 93.6 | 92.9 | 92.8 |
| *Proprietary Multimodal Models* | | | | | | | | | | | | | | | | | |
| Gemini 1.5 Pro Team et al. (2023) | 1fps | 87.3 | 67.0 | 80.2 | 54.5 | 68.3 | 67.4 | 70.8 | 68.6 | 75.7 | 52.7 | 62.3 | 35.5 | 74.2 | 61.7 | 57.2 | 65.3 |
| GPT-4o OpenAI (2024) | 64 | 69.1 | 65.1 | 65.5 | 50.0 | 68.3 | 63.7 | 63.6 | 49.8 | 71.0 | 55.4 | 58.7 | 27.6 | 73.2 | 59.4 | 53.4 | 58.6 |
| *Open-source Offline VideoLLMs* | | | | | | | | | | | | | | | | | |
| LLaVA-NeXT-Video-7B Zhang et al. (2024e) | 64 | 69.8 | 59.6 | 66.4 | 50.6 | 72.3 | 61.4 | 63.3 | 51.2 | 64.2 | 9.7 | 41.7 | 34.1 | 67.6 | 60.8 | 54.2 | 53.1 |
| LLaVA-OneVision-7B Li et al. (2024a) | 64 | 67.1 | 58.7 | 69.8 | 49.4 | 71.3 | 60.3 | 62.8 | 52.5 | 58.8 | 23.7 | 45.0 | 24.8 | 66.9 | 60.8 | 50.9 | 52.9 |
| Qwen2-VL-7B Wang et al. (2024a) | 64 | 69.1 | 53.2 | 63.8 | 50.6 | 66.3 | 60.9 | 60.7 | 44.4 | 66.9 | 34.4 | 48.6 | 30.1 | 65.7 | 50.8 | 48.9 | 52.7 |
| InternVL-V2-8B Chen et al. (2024b) | 64 | 68.5 | 58.7 | 69.0 | 44.9 | 67.3 | 56.0 | 60.7 | 43.1 | 61.5 | 27.4 | 44.0 | 25.8 | 57.6 | 52.9 | 45.4 | 50.1 |
| LongVU-7B Shen et al. (2024) | 1fps | 55.7 | 49.5 | 59.5 | 48.3 | 68.3 | 63.0 | 57.4 | 43.1 | 66.2 | 9.1 | 39.5 | 16.6 | 69.0 | 60.0 | 48.5 | 48.5 |
| *Open-source Online VideoLLMs* | | | | | | | | | | | | | | | | | |
| Flash-VStream-7B Zhang et al. (2024a) | 1fps | 25.5 | 32.1 | 29.3 | 33.7 | 29.7 | 28.8 | 29.9 | 36.4 | 33.8 | 5.9 | 25.4 | 5.4 | 67.3 | 60.0 | 44.2 | 33.2 |
| VideoLLM-online-8B Chen et al. (2024a) | 2fps | 8.1 | 23.9 | 12.1 | 14.0 | 45.5 | 21.2 | 20.8 | 22.2 | 18.8 | 12.2 | 17.7 | - | - | - | - | - |
| Dispider Qian et al. (2025) | 1fps | 57.7 | 49.5 | 62.1 | 44.9 | 61.4 | 51.6 | 54.5 | 48.5 | 55.4 | 4.3 | 36.1 | 18.0 | 37.4 | 48.8 | 34.7 | 41.8 |
| TimeChat-Online-7B Yao et al. (2025) | 1fps | 69.8 | 48.6 | 64.7 | 44.9 | 68.3 | 55.4 | 58.6 | 53.9 | 62.8 | 9.1 | 42.0 | 32.5 | 36.5 | 40.0 | 36.4 | 45.6 |
| **StreamAgent-7B (Ours)** | 1fps | **71.2** | 53.2 | 63.6 | 53.9 | 67.3 | 58.7 | 61.3 | 54.8 | 58.1 | 25.8 | 41.7 | 35.9 | 48.4 | 52.0 | 45.4 | **49.4** |

Table 1: Evaluation results on OVO-Bench comprising three categories: i) *Real-Time Visual Perception* (OCR: Optical Character Recognition, ACR: Action Recognition, ATR: Attribute Recognition, STU: Spatial Understanding, FPD: Future Prediction, OJR: Object Recognition), ii) *Backward Tracing* (EPM: Episodic Memory, ASI: Action Sequence Identification, HLD: Hallucination Detection), and iii) *Forward Active Responding* (REC: Repetition Event Count, SSR: Sequential Steps Recognition, CRR: Clues Reveal Responding).

threshold $\alpha$. The softmax-normalized importance of the $i$-th frame is given by:

$$\beta_i = \frac{e^{\mathbf{S}_{h,i}}}{\sum_{j=1}^{l} e^{\mathbf{S}_{h,j}}}, \tag{10}$$

where $\mathbf{S}_{h,i}$ denote the attention scores of $i$-th frame. If $\alpha = 6$ and the score of the $i$-th frame falls below $\max(\mathbf{S}_h) - 6$, its softmax weight $\beta_i$ is at most $1/e^6 \approx 1/403.4$ of the maximum. Such negligible weights justify its exclusion, contributing minimally to the final attention distribution.

• *Answering with Retrieved KV.* The retrieved KV-caches serve as the contextual input for video question-answering. As illustrated in Figure 3 (c), the KV entries corresponding to the current question are concatenated with the KV-caches retrieved from the long-term memory. In this case, the $K$ and $V$ used for attention computation are represented as:

$$K = [HX_{\text{out}}^k, X_q \cdot W_K], \quad V = [H_{\text{out}}^v, X_q \cdot W_V], \tag{11}$$

where $H_{\text{out}}^k$ and $H_{\text{out}}^v$ represent the key and value cache outputs which are retrieved from the long-term memory storage modules, respectively, and $X_q$ denotes either the current input question or the next output token that will be sequentially generated during the decoding inference process.

## 3 EXPERIMENTS

In this section, we comprehensively present our experimental studies. Section 3.1 details the experimental setup, including datasets, baselines, and necessary implementation settings. Section 3.2 reports the overall performance of our proposed model compared with various existing approaches. Section 3.3 further provides ablation studies to analyze the contribution of each individual component.

### 3.1 IMPLEMENTATION DETAILS

We employ the lightweight Qwen2.5VL-3B (Bai et al., 2025) as the agent's core for planning and tool coordination, and the larger Qwen2.5VL-7B (Bai et al., 2025) for precise interactions. Input video frames are resized to 224×224 at 1 FPS, following Dispider (Qian et al., 2025). For long-term memory, we first store the KV-cache in GPU memory, and offload to CPU memory once a predefined threshold is reached. The default value of $\alpha$ is set to 3. All experiments are conducted on NVIDIA A800 (80GB) GPUs with FP16 precision. Further setup details are provided in the Appendix C.

### 3.2 RESULTS ON STREAMING VIDEO BENCHMARKS

We evaluate StreamAgent on three streaming video benchmarks tailored to different aspects of streaming video understanding, including StreamingBench (Lin et al., 2024), OVO-Bench (Li et al., 2025d) and OVbench (Huang et al., 2025b), under strict real-time settings. Table 1 reports StreamAgent's performance on OVO-Bench which consists of 12 diverse tasks grouped into three core capabilities: Backward Tracing, Real-Time Visual Perception, and Forward Active Responding. StreamAgent achieves the highest overall performance among online models and even outperforms

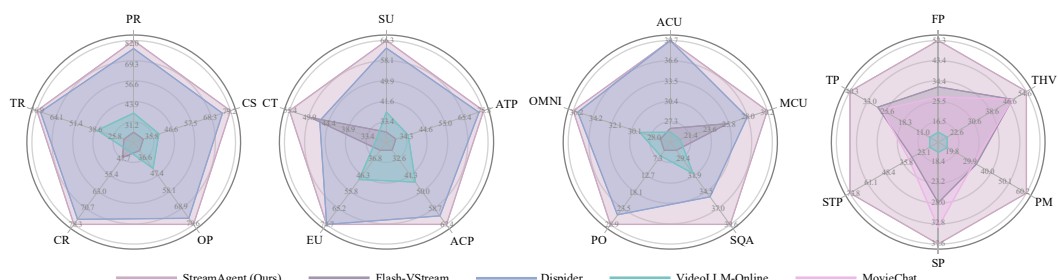

Figure 4: Comparative radar plots of StreamAgent and existing online video LLMs on diverse benchmarks, including StreamingBench (Lin et al., 2024) (first three) and OV-Bench (Huang et al., 2025b) (last).

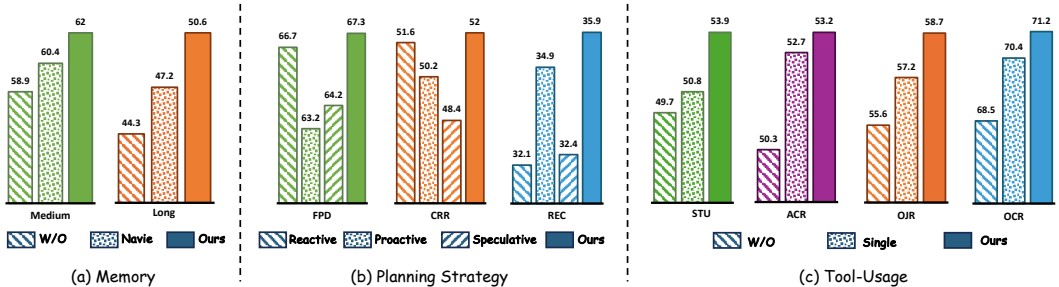

Figure 5: **Ablation Study** on three core components: (a) The effectiveness of the memory module is evaluated on the *medium* and *long* subsets of the VideoMME benchmark. (b) The impact of different planning strategies is evaluated on an OVOBench Li et al. (2025d) subset designed to test temporal foresight. (c) Tool usage is assessed on a separate subset to examine how proactive interaction enhances performance.

most offline. Notably, it improves Forward Active Responding by 10.7% over Dispider (Qian et al., 2025), and closely matches the top offline model, highlighting the effectiveness of proactive planning and streaming-aware design. To further assess generalization, we evaluate on StreamingBench and OVBench. While StreamingBench focuses on real-time perception with fine-grained, time-stamped queries, OVBench emphasizes long-range temporal reasoning across 16 subtasks spanning past, present, and future. As illustrated in Figure 4, StreamAgent achieves state-of-the-art results on both benchmarks, significantly outperforming previous online models and approaching offline baselines. These gains are driven by our proactive StreamAgent, which accurately predicts task-relevant temporal and spatial cues, and our efficient streaming KV-cache mechanism, which enhances long-form understanding by selective recall while reducing computational overhead.

## 3.3 ABLATION STUDY

**Effectiveness of Agent Design.** To evaluate the contribution of each core component in our StreamAgent framework, we conduct a comprehensive ablation study along three key axes: *incremental memory, planning strategy, and tool-based active perception*. Each experiment is designed to isolate the effect of the corresponding module on the final performance.

*1. Memory.* Specifically, we compare the performance of our memory mechanism against approaches without memory and those using traditional segmentation-based captioning, where captions are generated for each segment and then concatenated. Our method shows significant gains on longer videos ( Figure 5 (a)), which confirms that Markov memory not only reduces redundancy but also enhances long-term context modeling crucial for high-quality video question answering.

*2. Planning.* For planning, we compare *Reactive, Proactive, and Speculative.* strategies with our heuristic-based planning. As shown in Figure 5 (b), our approach consistently outperforms fixed strategies across all time scales. Notably, the performance of speculative planning remains relatively limited across videos of all lengths, likely due to the inherent uncertainty of long-horizon predictions. Additionally, the nature of the dataset may contribute to this trend, as many questions tend to favor short-term reasoning and do not require extensive temporal foresight.

*3. Tool-Use.* Finally, we ablate external tools to evaluate the impact of proactive information hunting. As shown in Figure 5 (c), tool-based perception significantly improves accuracy with minimal overhead, underscoring the value of targeted, strategic perception beyond passive observation.

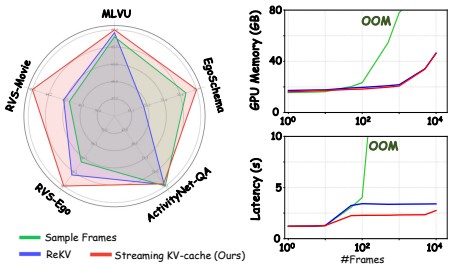

Figure 6: Streaming KV-cache boosts Streaming VideoQA accuracy and achieves over 30% faster inference than ReKV when evaluated with Qwen2.5vl-7B on an A800 (80GB).

Figure 7: Results on offline long video benchmarks. We report the accuracy on the MLVU, LongVideobench and VideoMME without subtitles.

| Model | #Frames | MLVU | LongVideoBench | VideoMME | |
|---|---|---|---|---|---|
| | | | | overall | long |
| **Video Length** | - | 3~120 min | 8 sec~60 min | 1~60 min | 30~60 min |
| **Open-Source Offline VideoLLMs** | | | | | |
| LLaMA-VID-7B | 1fps | 33.2 | - | - | - |
| MovieChat-7B | 2048 | 25.8 | - | 38.2 | 33.4 |
| LLaVA-Next-Video-7B | 32 | - | 43.5 | 46.6 | - |
| VideoChat2-7B | 16 | 47.9 | 39.3 | 39.5 | 33.2 |
| LongVA-7B | 128 | 56.3 | - | 52.6 | 46.2 |
| Qwen2.5-VL-7B | 1fps | 66.9 | 61.5 | 63.2 | 50.4 |
| **Open-source Online VideoLLMs** | | | | | |
| Dispider-7B | 1fps | 61.7 | - | 57.2 | - |
| VideoChat-Online-8B | 2fps | - | - | 52.8 | 44.9 |
| TimeChat-Online-7B | 1fps | 65.4 | 57.7 | 62.5 | 49.2 |
| **StreamAgent-7B** | 1fps | **67.2** | **57.9** | **62.9** | **50.6** |

**Effectiveness of Streaming KV-cache.** To evaluate the effectiveness of our streaming KV-cache mechanism, we follow the experimental protocol of ReKV (Di et al., 2025) and test on five diverse benchmarks covering different kinds of traditional scenarios: *MLVU* (Zhou et al., 2024), *EgoSchema* (Mangalam et al., 2023), *ActivityNet-QA* (Yu et al., 2019), *RVS-Ego* and *RVS-Movie* (Zhang et al., 2024a). We compare against representative baselines: Sampled Frames, and ReKV. As shown in Figure 6, our hybrid architecture achieves up to 30% lower latency than ReKV as frame counts increase, while maintaining GPU memory usage comparable to Sampled Frames. Moreover, it consistently outperforms all baselines. Further details are provided in Appendix C.

**Ablation Study on Video Understanding Agent.** In the offline setting, we further compare the performance of various video understanding agents, VideoAgent (Wang et al., 2024b), VideoMemAgent (Fan et al., 2024) and ReAgent-V (Zhou et al., 2025a).

| Model | #Frames | MLVU | VideoMME | | | |
|---|---|---|---|---|---|---|
| | | | Short | Medium | Long | Overall |
| VideoAgent | 87 | 57.8 | 63.6 | 55.4 | 49.0 | 56.0 |
| VideoMemAgent | 72 | 58.2 | 55.3 | **64.2** | **52.7** | 57.4 |
| ReAgent-V | 35 | 60.7 | **73.5** | 58.2 | 49.8 | 60.7 |
| **StreamAgent-7B** | 1fps | **67.2** | 73.4 | 62.0 | 50.6 | **62.9** |

Table 2: Ablation study of different video understanding agents.

Benefiting from our memory mechanism and streaming KV-cache, our method processes videos at 1 FPS and shows outperformance (Table 2) on video benchmarks.

**Results on Offline Long Video Tasks.** In addition to the streaming setting, we also evaluate StreamAgent on three offline long-form video understanding benchmarks: VideoMME (Fu et al., 2024), MLVU (Zhou et al., 2024), and LongVideoBench (Wu et al., 2024b). In the offline setting, the entire video is provided as input to the VideoLLMs. Table 7 demonstrates that StreamAgent exhibits superior offline video understanding capabilities compared to recent state-of-the-art VideoLLMs, including LLaMA-VID (Li et al., 2024b), MovieChat (Song et al., 2024), LLava-next-video (Zhang et al., 2024e), VideoChat (Li et al., 2023), LongVA (Zhang et al., 2024d), Qwen2.5-VL (Wang et al., 2024a), Dispider (Qian et al., 2025), VideoChat-Online (Huang et al., 2025b). Thanks to our incremental memory module and key-value cache retrieval mechanism, StreamAgent handles this conventional scenario well and achieves competitive performance with offline models.

**Case Study.** To evaluate StreamAgent's capability in *Proactive Event Anticipation*, we present a case form streamingBench (Lin et al., 2024) dataset involving the question "What was the total number of goals scored in the match?"–a scene from a football match, as shown in Figure 8. When the model lacked a plan, it prematurely triggered a response after the first goal and incorrectly answered "1 goal." Through heuristic planning, we avoid such premature answers by using tools to focus on key signals and responding after those signals are observed. This case highlights the value of future-aware planning and tool integration in proactive video understanding.

# 4 RELATED WORK

**Streaming Video Understanding** demands real-time responses to user queries, even as video durations extend indefinitely. This stands in contrast to traditional video understanding benchmarks (Carreira & Zisserman, 2017; Alamri et al., 2019; Mangalam et al., 2023), which rely on uniformly sampled clips for tasks like action recognition, multi-turn dialogue, or egocentric question answering. To this end, recent works (Zhang et al., 2024a; Chen et al., 2024a; Liu et al., 2025a; Chen et al., 2025b) have explored online models that incrementally process current and past frames.

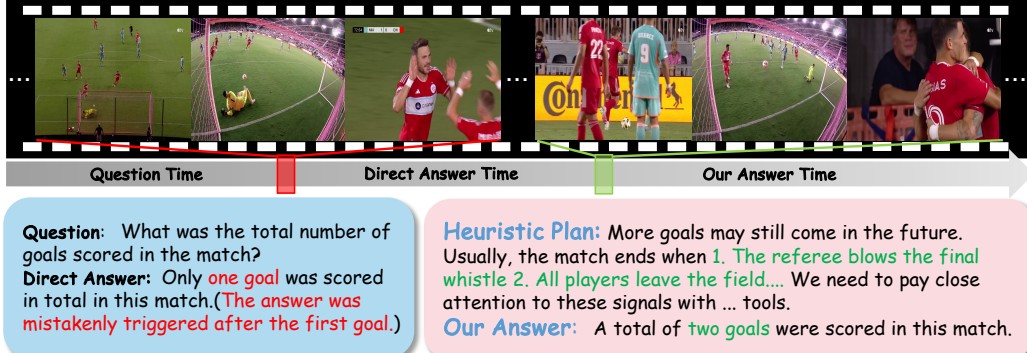

Figure 8: **Proactive Anticipation examples.** The video stream is processed frame by frame. On the left, a Direct Answer is triggered prematurely after the first goal, leading to an incorrect response. On the right, the proposed Heuristic Plan waits for clearer signs, such as the referee's whistle, before answering.

However, real-world applications (*e.g.,* autonomous driving) demand streaming video understanding with continuous perception (Wu et al., 2024c; Zhang et al., 2024c; Guo et al., 2025), efficient information extraction, proactive decision-making, and responsive interaction under dynamically evolving visual content, even without explicit queries. Recent works (Wang et al., 2025a; Yang et al., 2025d; Wu et al., 2024a) such as VideoLLM-Online (Chen et al., 2024a), VideoStream (Qian et al., 2024), Flash-VStream (Zhang et al., 2024a), and Dispider (Qian et al., 2025) have explored the understanding of streaming video from different perspectives, including streaming dialogue frameworks (Chatterjee et al., 2025; Xiong et al., 2025; Li et al., 2025b), memory-efficient encoding, and disentangled perception-decision-reaction pipelines. In contrast, we adopt a streaming agent that integrates continuous perception with task-driven planning to anticipate future task-relevant information for proactive and efficient video understanding.

**Agentic Framework for Video Understanding** leveraging the powerful capabilities of MLLM, recent studies have explored agentic approaches to tackle the complex challenges of video understanding (Zhi et al., 2025; Kugo et al., 2025; Yuan et al., 2025; Chen et al., 2025a; Jeoung et al., 2024; Zhang et al., 2024b). However, these approaches rely on sliding window inference or dense retrieval, resulting in two major limitations: high computational overhead due to exhaustive processing and lack of explicit anticipation, thereby hindering proactive decisions in dynamic real-time scenes. Therefore, this work addresses streaming video understanding, which integrates task-driven planning and future anticipation for efficient, accurate, and robust perception and response in continuous video streams.

**Memory Optimization for Video Understanding** Video understanding tasks often incur substantial memory overhead, primarily due to the storage demands of the KV-cache, and recent research has explored various strategies to mitigate this issue. LiveVLM Ning et al. (2025) and StreamMem Yang et al. (2025c) reduce memory consumption by compressing the KV-cache. Flash_VStream Zhang et al. (2024a) alleviates memory pressure by compressing visual features through the STAR mechanism. In this work, we retain retrieval capability while employing a dynamic attention pattern that adaptively selects informative tokens to enhance long-range interaction and fine-grained detail capture.

## 5 CONCLUSION

In this paper, we propose StreamAgent for streaming video understanding, which proactively anticipates the temporal intervals and spatial regions of future task-relevant information, enabling proactive and accurate responses. Furthermore, we introduce streaming KV-cache that constructs a hierarchical memory structure and supports selective recall of relevant tokens for efficient semantic retrieval. Experiments demonstrate that our method achieves state-of-the-art performance on streaming and long video understanding benchmarks in both response accuracy and real-time efficiency.

**Limitation:** We clarify the limitations of our proposed StreamAgent: *(i):* StreamAgent does not explicitly model every type of video reasoning scenario. This is understandable since the framework prioritizes real-time efficiency over exhaustive coverage, and thus may underperform in niche or artificially complex cases. *(ii):* Although tool-augmented planning improves proactive responses, the extent to which different tool choices influence outcomes is not fully explored. Additional experiments may be needed to better understand this interaction, but it does not hinder applicability.

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

# APPENDIX

## ABSTRACT

Due to the space constraints of the main manuscript, this Supplementary Material provides a comprehensive presentation of additional details and analyses. The content is structured as follows: (1) Additional Related Work. (2) A detailed overview of the datasets and methods, including streaming and offline settings. (3) Analysis of Memory Uasage of Streaming KV cache. (4) A detailed explanation of Detail of Evaluation. (5) Pseudocode of StreamAgent and Streaming KV cache. (6)Details of Prompt in StreamAgent. (7) Limitation. (8) Future Work.

## APPENDIX TABLE OF CONTENTS

## A   ADDITIONAL RELATED WORK.

**Long Video Understanding** Inspired by the powerful reasoning capabilities of LLMs Yu et al. (2025), recent works have explored using LLMs to address complex video-related tasks. Since LLMs primarily process text, various methods (Korbar et al., 2024; Weng et al., 2024; Zhang et al., 2023; Jin et al., 2023; Wang et al., 2024d) have been developed to efficiently train multimodal projectors to connect the visual encoder and LLMs Hu et al. (2024a); Tang et al. (2025); Liu et al. (2025c); Xue et al. (2025); Liu et al. (2025c) or leverage caption-centric information. With the success of VideoLLM Team et al. (2025b); Shu et al. (2025); Yan et al. (2025); Team et al. (2025a), attention has shifted to the more complex task of long video understanding Wang et al. (2025b); Zhou et al. (2025b); Wang et al. (2024e), which is typically facilitated through token compression Pollard & Wray (2025); Lee et al. (2024a); Choi et al. (2024) and frame selection Liu et al. (2025b); Zhang et al. (2025); Hu et al. (2025); Huang et al. (2025a). This approach effectively reduces the token sequence length, allowing for better handling of longer videos. Recent work has enhanced the model's capabilities by using inference Yang et al. (2025a); Feng et al. (2025); Li et al. (2024c); Fei et al. (2024); Tian et al. (2025); Wu et al. (2025a), often searching through long videos to obtain more granular details and achieve more accurate grounding during video understanding.

**Memory Optimization for Video Understanding** Video understanding tasks often incur substantial memory overhead, primarily due to the storage demands of the KV-cache, and recent research has explored various strategies to mitigate this issue. LiveVLM Ning et al. (2025) and StreamMem Yang et al. (2025c) reduces memory consumption by compressing the KV-cache. Flash_VStream Zhang et al. (2024a) alleviates memory pressure by compressing visual features through the STAR mechanism. TimeChat-Online Yao et al. (2025), InfiniPot-V Kim et al. (2025) and EgoPrune Li et al. (2025a) alleviates memory pressure by reducing visually redundant tokens in videos. VideoScan Li et al. (2025b) and StreamMind Ding et al. (2025) represents each video frame with a single token, drastically reducing the length of the visual token sequence. QuickVideo Schneider et al. (2025) discards tokens deemed unimportant based on accumulated attention scores, retaining only a fixed number of tokens (e.g., Top-K). ReKV Di et al. (2025) extends this idea by incorporating a retrieval mechanism, enabling access to historical information that would otherwise be discarded. However, both methods rely on static attention patterns to select key visual tokens. In this work, we preserve the

retrieval capability while adopting a dynamic attention pattern, which adaptively selects informative visual tokens, thereby improving the model's ability to capture long-range interactions and nuanced visual details.

## B  BENCHMARK AND BASELINE.

**MLVU**$_{\text{dev-mc}}$ (Zhou et al., 2024) is the multiple-choice subset of the MLVU-dev benchmark. It focuses on evaluating the long-form video understanding of MLMs. The question-answer pairs are manually labeled and can be divided into 3 groups: single-detail, multi-detail, and holistic. The evaluation metric is Accuracy.

**LongVideoBench** (Wu et al., 2024b) is a video question-answering benchmark specifically designed for LMMs. It aims to evaluate LMMs' ability to process long, interleaved video and subtitle inputs up to an hour in length. The benchmark includes: 3,763 varying-length web-collected videos with their subtitles, covering diverse themes. A novel "referring reasoning" video question-answering task, where questions contain referring queries that point to specific video contexts, requiring the model to reason over relevant video details. 6,678 human-annotated multiple-choice questions, categorized into 17 fine-grained categories.

**VideoMME** (Fu et al., 2024) is the first comprehensive, full-spectrum evaluation benchmark for LMMS specifically designed for video analysis. It addresses the current gap in assessing LMMs' ability to process sequential visual data, moving beyond their traditional focus on static image understanding. Here are its key features: Diverse Video Types: Video-MME covers 6 primary visual domains and 30 subfields, ensuring broad applicability across various scenarios. Varied Temporal Durations: It includes short, medium, and long videos, ranging from 11 seconds to 1 hour, to evaluate robust contextual understanding. Broad Data Modalities: Beyond video frames, the benchmark integrates other modalities like subtitles and audio to assess the all-around capabilities of MLMs. High-Quality Annotations: It uses rigorous manual labeling by expert annotators to ensure precise and reliable model assessment.

**EgoSchema** (Mangalam et al., 2023) is a diagnostic benchmark for long VideoQA, featuring over 5000 multiple-choice questions and long temporal certificate length. It challenges AI models with long-term understanding, as current state-of-the-art models achieve significantly lower accuracy compared to human performance.

**ActivityNet-QA** (Yu et al., 2019) encompasses human-annotated QA pairs on 5,800 videos derived from the ActivityNet (Caba Heilbron et al., 2015) dataset. This benchmark is designed to assess the capabilities of VideoQA models in long-term spatiotemporal reasoning. Our evaluation methodology aligns with that of Video-ChatGPT (Maaz et al., 2024), employing `GPT-3.5-turbo-0613` to judge the accuracy of the open-ended VideoQA responses.

**RVS-Ego** and **RVS-Movie** (Zhang et al., 2024a) are Streaming VideoQA benchmarks, constructed using 10 long videos from the Ego4D dataset (Grauman et al., 2022) and 22 long videos from the MovieNet dataset (Huang et al., 2020), respectively. These benchmarks feature open-ended questions paired with timestamps, which are initially generated by GPT-4V (OpenAI, 2023b) and GPT-4 (OpenAI, 2023a), and subsequently refined through manual filtering.

**OVO-Bench** (Li et al., 2025d) is a dataset specifically designed to evaluate Video Large Language Models (Video-LLMs) on their ability to understand online, streaming video content. Unlike traditional offline benchmarks that assess models after they have seen the entire video, OVO-Bench focuses on real-time reasoning, requiring models to answer questions at any given timestamp while the video is playing. OVO-Bench defines a comprehensive suite of tasks to evaluate the temporal reasoning and visual understanding capabilities of Video Large Language Models (Video-LLMs) in online settings. These tasks are grouped into three core categories. The first, Backward Tracing, assesses a model's ability to recall and reason about past events, including retrieving key moments (EPM), identifying the correct sequence of actions (ASI), and detecting hallucinated responses to irrelevant questions (HLD). The second category, Real-Time Visual Perception, focuses on understanding the present visual context through spatial reasoning (STU), object recognition (OJR), attribute identification (ATR), action recognition (ACR), text recognition (OCR), and even predicting what might happen next (FPD). The third category, Forward Active Responding, goes beyond passive perception by requiring models to delay responses until sufficient evidence is available—such as recognizing repeated events

| Model | Params | Frames | Real-Time Visual Understanding | | | | | | | | | | | Omni-Source Understanding | | | | | Contextual Understanding | | | | | Overall |
|---|---|---|---|---|---|---|---|---|---|---|---|---|---|---|---|---|---|---|---|---|---|---|---|---|
| | | | OP | CR | CS | ATP | EU | TR | PR | SU | ACP | CT | All | ER | SCU | SD | MA | All | ACU | MCU | SQA | PO | All | |
| **Human** | | | | | | | | | | | | | | | | | | | | | | | | |
| Human‡ | - | - | 89.47 | 92.00 | 93.60 | 91.47 | 95.65 | 92.52 | 88.00 | 88.75 | 89.74 | 91.30 | 91.46 | 88.00 | 88.24 | 93.60 | 90.27 | 90.26 | 88.80 | 90.40 | 95.00 | 100 | 93.55 | 91.66 |
| **Proprietary MLLMs** | | | | | | | | | | | | | | | | | | | | | | | | |
| Gemini 1.5 pro | - | 1 fps | 79.02 | 80.47 | 83.54 | 79.67 | 80.00 | 84.74 | 77.78 | 64.23 | 71.95 | 48.70 | 75.69 | 46.80 | 39.60 | 74.90 | 80.00 | 60.22 | 51.41 | 40.73 | 54.80 | 45.10 | 48.73 | 67.07 |
| GPT-4o | - | 64 | 77.11 | 80.47 | 83.91 | 76.47 | 70.19 | 83.80 | 66.67 | 62.19 | 69.12 | 49.22 | 73.28 | 41.20 | 37.20 | 43.60 | 56.00 | 44.50 | 41.20 | 38.40 | 32.80 | 56.86 | 38.70 | 60.15 |
| Claude 3.5 Sonnet | - | 20 | 80.49 | 77.34 | 82.02 | 81.73 | 72.33 | 75.39 | 61.11 | 61.79 | 69.32 | 43.09 | 72.44 | 31.60 | 34.00 | 32.80 | 48.80 | 36.80 | 38.40 | 34.80 | 34.40 | 64.71 | 37.70 | 57.68 |
| **Open-Source Video MLLMs** | | | | | | | | | | | | | | | | | | | | | | | | |
| LLaVA-OneVision | 7B | 32 | 80.38 | 74.22 | 76.03 | 80.72 | 72.67 | 71.65 | 67.59 | 65.45 | 65.72 | 45.08 | 71.12 | 40.80 | 37.20 | 33.60 | 44.80 | 38.40 | 35.60 | 36.00 | 27.27 | 29.55 | 32.74 | 56.36 |
| Qwen2-VL | 7B | 0.2-1 fps | 75.20 | 82.81 | 73.19 | 77.45 | 68.32 | 71.03 | 72.22 | 61.19 | 61.47 | 46.11 | 69.04 | 41.20 | 22.00 | 32.80 | 43.60 | 34.90 | 31.20 | 26.00 | 39.60 | 22.73 | 31.66 | 54.14 |
| MiniCPM-V 2.6 | 8B | 32 | 71.93 | 71.09 | 77.92 | 75.82 | 64.60 | 65.73 | 70.37 | 56.10 | 62.32 | 53.37 | 67.44 | 40.80 | 24.00 | 34.00 | 41.20 | 35.00 | 34.00 | 31.60 | 41.92 | 22.22 | 34.97 | 53.85 |
| LLaVA-NeXT-Video | 32B | 64 | 78.20 | 70.31 | 73.82 | 76.80 | 63.35 | 69.78 | 57.41 | 56.10 | 64.31 | 38.86 | 66.96 | 37.69 | 24.80 | 34.40 | 42.80 | 34.90 | 29.20 | 30.40 | 35.35 | 18.18 | 30.79 | 52.77 |
| InternVL-V2 | 8B | 16 | 68.12 | 60.94 | 69.40 | 77.12 | 67.70 | 62.93 | 59.26 | 53.25 | 54.96 | 56.48 | 63.72 | 37.60 | 26.40 | 37.20 | 42.00 | 35.80 | 32.00 | 31.20 | 32.32 | 40.91 | 32.42 | 51.40 |
| Kangaroo | 7B | 64 | 71.12 | 84.38 | 70.66 | 73.20 | 67.08 | 61.68 | 56.48 | 55.69 | 62.04 | 38.86 | 64.60 | 37.60 | 31.20 | 28.80 | 39.20 | 34.20 | 32.80 | 26.40 | 33.84 | 16.00 | 30.06 | 51.10 |
| LongVA | 7B | 128 | 70.03 | 63.28 | 61.20 | 70.92 | 62.73 | 59.50 | 61.11 | 53.66 | 54.67 | 34.72 | 59.96 | 39.60 | 32.40 | 28.00 | 41.60 | 35.40 | 32.80 | 29.60 | 30.30 | 15.91 | 29.95 | 48.66 |
| VILA-1.5 | 8B | 14 | 53.68 | 49.22 | 70.98 | 56.86 | 53.42 | 53.89 | 54.63 | 48.78 | 50.14 | 17.62 | 52.32 | 41.60 | 26.40 | 28.40 | 36.00 | 33.10 | 26.80 | 34.00 | 23.23 | 17.65 | 27.35 | 43.20 |
| Video-CCAM | 14B | 96 | 56.40 | 57.81 | 65.30 | 62.75 | 64.60 | 51.40 | 42.59 | 47.97 | 49.58 | 31.61 | 53.96 | 33.60 | 22.00 | 28.40 | 34.80 | 29.70 | 27.60 | 24.40 | 16.67 | 22.73 | 22.88 | 42.53 |
| Video-LLaMA2 | 7B | 32 | 55.86 | 55.47 | 57.41 | 58.17 | 52.80 | 43.61 | 39.81 | 42.68 | 45.61 | 35.23 | 49.52 | 30.40 | 32.40 | 30.40 | 36.00 | 32.40 | 24.80 | 26.80 | 18.67 | 0.00 | 21.93 | 40.40 |
| **Streaming MLLMs** | | | | | | | | | | | | | | | | | | | | | | | | |
| Flash-VStream | 7B | - | 25.89 | 43.57 | 24.91 | 23.87 | 27.33 | 13.08 | 18.52 | 25.20 | 23.87 | 48.70 | 23.23 | 25.91 | 24.90 | 25.60 | 28.40 | 26.00 | 24.80 | 25.20 | 26.80 | 1.96 | 24.12 | 24.04 |
| VideoLLM-online | 8B | 2 fps | 39.07 | 40.06 | 34.49 | 31.05 | 45.96 | 32.40 | 31.48 | 34.16 | 42.49 | 27.89 | 35.99 | 31.20 | **26.51** | 24.10 | 32.00 | 28.45 | 24.19 | 29.20 | 30.80 | 3.92 | 26.55 | 32.48 |
| Dispider | 7B | 1 fps | 74.92 | 75.53 | 74.10 | 73.08 | 74.44 | 59.92 | 76.14 | 62.91 | 62.16 | 45.80 | 67.63 | 35.46 | 25.26 | 38.57 | 43.34 | 35.66 | 39.62 | 27.65 | 34.80 | 25.34 | 33.61 | 53.12 |
| **StreamAgent** | 7B | 1 fps | **79.63** | **78.31** | **79.28** | **75.87** | **74.74** | **76.92** | **82.94** | **66.31** | **73.69** | **55.40** | **74.28** | **35.86** | 26.26 | **38.87** | **44.04** | **36.26** | **39.72** | **30.25** | **39.60** | **28.90** | **34.62** | **57.02** |

Table 3: Performance comparison on StreamingBench on Omni-source Understanding, Contextual Understanding, and Real-Time Visual Understanding. Omni-source Understanding includes Emotion Recognition (ER), Scene Understanding (SCU), Source Discrimination (SD), and Multimodal Alignment (MA). Contextual Understanding includes Misleading Context Understanding (MCU), Anomaly Context Understanding (ACU), Sequential Question Answering (SQA) and Proactive Output (PO). Real-Time Visual Understanding includes Object Perception (OP), Causal Reasoning (CR), Clips Summarization (CS), Attribute Perception (ATP), Event Understanding (EU), Text-Rich Understanding (TR), Prospective Reasoning (PR), Spatial Understanding (SU), Action Perception (ACP), and Counting (CT). Results are categorized into Human, Proprietary MLLMs, and Open-Source MLLMs for a comprehensive evaluation.

(REC), detecting procedural transitions (SSR), and waiting for critical clues before answering (CRR). Together, these tasks provide a rigorous evaluation of how well a model can perceive, remember, and anticipate in dynamic, real-world video environments.

**StreamingBench** (Lin et al., 2024) introduces the first comprehensive benchmark tailored for evaluating how well multimodal large language models (MLLMs) understand streaming video in realistic scenarios. Comprising 900 videos with 4,500 human-curated QA pairs spread across 18 task types, it simulates continuous video interactions by posing five time-staggered questions per video. The benchmark assesses three key understanding dimensions: real-time visual understanding (recognizing objects, actions, or text in the moment), omni-source understanding (integrating synchronized visual and audio inputs), and contextual understanding (maintaining continuity across interactions, detecting anomalies, filtering misleading cues, and responding proactively based on preceding context).

**OVBench** (Huang et al., 2025b) are Streaming VideoQA benchmarks, denoting the unique challenges and characteristics of online video understanding compared to traditional offline settings. First, it emphasizes the online temporal perspective, where questions are grounded in specific moments, past, present, or future, enabling much finer time sensitivity than offline approaches. It highlights how time-dependent contexts cause answers to evolve dynamically, meaning that the same question may yield different responses as the stream progresses. Third, it points to the need for real-time spatio-temporal interaction in applications like AR glasses or autonomous driving, where immediate and accurate responses to live environments are crucial.

To rigorously assess the performance of StreamAgent, we compare it against a broad array of baseline models that span different capabilities and design philosophies. Proprietary closed-source models include GPT-4o, known for its advanced multimodal reasoning, and Gemini-1.5-Pro, optimized for long-context multimodal input. Among open-source video-language models, the evaluation includes:

**VideoChat2**: A chat-centric video-language model built to support multi-turn, conversational interactions grounded in video content. It emphasizes interactive understanding, with capabilities for dialogue continuity, temporal grounding, and multi-modal alignment, making it ideal for assistive agents and education scenarios.

**MiniCPM-V**: MiniCPM-V is a lightweight, vision-language model developed for efficient multi-modal understanding and generation tasks. Designed with a compact architecture, it achieves strong performance on image captioning, visual question answering, and other vision-language benchmarks, while maintaining low computational requirements. This makes MiniCPM-V ideal for deployment in resource-constrained environments such as mobile devices and edge computing scenarios.

**VILA-1.5**: VILA-1.5 is a state-of-the-art vision-language model that excels in multimodal understanding and generation tasks. It integrates visual and textual information through a unified architecture, enabling high performance on benchmarks such as image captioning, visual question answering, and image-text retrieval. With improved alignment between visual and linguistic modalities, VILA-1.5 offers both accuracy and efficiency, making it suitable for a wide range of real-world applications.

**Qwen2** and **Qwen2.5-VL** families: Developed by Alibaba, these models exhibit high performance on both image and video understanding benchmarks through advanced multi-modal alignment. Qwen2.5-VL particularly excels in handling long-context and dense visual-textual reasoning tasks, supported by a unified visual-language architecture.

**InternVL-2**: A high-performing open-source model that scales both the architecture and training data to improve generalization. It incorporates vision-language alignment techniques, dense region grounding, and optimized pretraining routines to support both short and long video tasks with high efficiency.

**Kangaroo**: Designed specifically for long-context video input, Kangaroo adopts hierarchical attention and memory-efficient tokenization strategies to process hundreds of frames. It is particularly effective for document-level video understanding, such as meeting summarization or sports analysis.

**Long-LLaVA**: A variant of LLaVA optimized for efficient multi-frame processing. It integrates temporal coherence constraints and cross-frame attention mechanisms, making it capable of capturing nuanced motion patterns and temporal dependencies for improved video QA and description.

**LLaVA-OneVision**: LLaVA-OneVision is an open-source large multimodal model that excels across single-image, multi-image, and video understanding tasks. Built on a SigLIP vision encoder and Qwen-2 language backbone, it uses a unified visual token strategy—including AnyRes-9 for high-res images and frame-level pooling for video—to balance representation across modalities.

**MovieChat**: MovieChat is an innovative long-video understanding framework that combines vision foundation models and large language models using a dual-memory mechanism inspired by the Atkinson–Shiffrin model. It processes video frames with a sliding-window tokenization approach and maintains a rapidly updated short-term memory buffer along with a compact long-term memory.

**LLaVA-NeXT-Video**: LLaVA-NeXT-Video is an open-source video-language assistant built by fine-tuning the LLaVA-NeXT image model (based on Qwen-1.5 32B) with high-quality video instruction data (830K samples) mixed with image data. It leverages the AnyRes representation—originally designed for high-resolution images—to naturally process videos as sequences of frames, enabling strong zero-shot video understanding ([huggingface.co][2]). With techniques like length generalization and DPO fine-tuning, the model achieves top open-source performance on benchmarks like Video-MME and NextQA-MC, while remaining efficient in inference and deployment.

**ReAgent-V**: ReAgent-V is an agent-based video-language framework that enhances large vision-language models with dynamic frame selection, multi-step reasoning, and self-correction. By integrating tool-using agents and a feedback-driven refinement process, it achieves superior performance on video understanding tasks with improved efficiency and accuracy.

**VideoAgent**:An agent framework for long-form video understanding that mimics human cognition through LLM-guided reasoning, CLIP-based retrieval, and VLM-driven state updates.

**VideoMemAgent**: An agent framework for structured long-form video understanding that integrates unified temporal and object memory with tool-augmented reasoning, enabling multi-round chain-of-thought inference across complex video content.

## C   MEMORY ANALYSIS OF STREAMING KV CACHE

In this section, we analyze the memory consumption of our approach, focusing on activation memory and key-value (KV) cache memory.

**Activation Memory Analysis.** The activation memory of modern LLM architecture mainly comes from two components of each transformer layer: 1) Attention layer and 2) MLP layer. We analyze the potential activation memory usage in formulas in the followings and show that chunked-prefill can approximately reduce the activation memory by $C$ times, where $C$ is the number of chunks.

***Attention Layer.*** For an inference computation, the input tensor of the $i$-th attention layer ($X^i$) is first layer-normalized to produce $X_a^i$, which is then fed into the attention module. The output of the attention module after a residual add and layer normalization forms the $X_O^i$. The $X_O^i$ can be calculated by:

$$Q^i = X_a^i \cdot W_Q^i, \quad K^i = X_a^i \cdot W_K^i, \quad V^i = X_a^i \cdot W_V^i \tag{12}$$

$$X_O^i = f_{\text{norm}} \left( f_{\text{Softmax}} \left( \frac{Q^i K^{iT}}{\sqrt{h}} \right) \cdot V^i \cdot w_O^i + X^i \right) \tag{13}$$

For simplicity, we ignore memory fragmentation; analyzing the computational data flow of the attention layer (Eq. xxx), the total activation memory with half precision can be expressed as:

$$\mathcal{M}_{\text{attn}} \approx (2B \cdot S \cdot n_h \cdot d_{\text{head}} + 2B \cdot S \cdot n_{kv} \cdot d_{\text{head}}) \cdot 2 \text{ bytes} \tag{4}$$

The first term accounts for storing the input hidden_states $X^i$ and **Q** tensors (Assume that the model's hidden dimension is equal to $n_h \times d_{head}$), while the second term accounts for the $K$ and $V$ tensors (i.e., KV caches). Assuming $B = 1$, $S \approx 921600$, $d_{\text{model}} = 3584$, $d_{\text{head}} = 128$, $n_h = 28$, $n_{kv} = 4$, we compute:

$$\mathcal{M}_{\text{attn}} = (2 \cdot 921600 \cdot 28 \cdot 128 + 2 \cdot 921600 \cdot 4 \cdot 128) \cdot 2 \tag{14}$$

$$= 15,099,494,400 \text{ bytes} \tag{15}$$

$$\approx \boxed{14.1 \text{ GB}} \tag{16}$$

With chunked-prefill using $C = 4096$ chunk size, we can reduce the sequence length $S$ by $C = 225$ times, reducing $\mathcal{M}_{\text{attn}}$ from 17.6 GB to approximately 0.08 GB. This dramatic reduction enables the processing of extremely long sequences that would otherwise be infeasible.

***MLP Layer.*** The SwiGLU (Swish-Gated Linear Unit) enhances transformer models through improved gating mechanisms and has been adopted as the default MLP architecture in many popular LLMs including InternVL2.5 and Qwen2.5 series. For input tensor $X^i$, the SwiGLU operation is defined as:

$$X^{i+1} = f_{\text{silu}} \left( X_O^i \cdot W_1 \right) * \left( W_2 \cdot X_O^i \right) \cdot W_3 + X_O^i \tag{17}$$

For a batch of sequences, activation memory analysis reveals requirements at each computational step. With batch size $B$, sequence length $S$, hidden dimension $d_{\text{model}}$, intermediate dimension $d_{\text{ff}}$, and data type float16, the total activation memory for a single SwiGLU layer is:

$$\mathcal{M}_{\text{act}} = (B \cdot S \cdot (2d_{\text{model}} + 3d_{\text{ff}})) \cdot 2 \text{ bytes} \tag{9}$$

For a one-hour video sampled with 1 FPS (3600 frames in total), parameters can be set $B = 1$, $S \approx 921600$, $d_{\text{model}} = 4096$, and $d_{\text{ff}} = 14336$:

$$\mathcal{M}_{\text{act}} = (1 \cdot 921600 \cdot (2 \cdot 4096 + 3 \cdot 14336)) \cdot 2 \tag{18}$$

$$= 94,371,840,000 \text{ bytes} \tag{19}$$

$$\approx \boxed{87.9 \text{ GB}} \tag{20}$$

| Hyper-parameter | Value |
|---|---|
| *Visual Encoder* | |
| Frame Sampling Rate | FPS=1 |
| Input Resolution | 224*224 |
| Visual Tokens per Image | 32 |
| Patch Size | 14x14 |
| *Large Language Model* | |
| Number of Layers | 28 |
| Hidden Size | 3584 |
| Vocabulary Size | 152064 |
| Number of Attention Heads | 28 |
| Number of KV Heads | 4 |
| *Streaming KV-cache* | |
| $\alpha$ | 3 |
| Maximum number of retrievals | 256 |
| GPU-Memory(A100 SXM4) | 80GB |

Table 4: The implementation settings of the evaluation.

This substantial memory requirement highlights the computational challenges in deploying SwiGLU-based models for high-resolution inputs with extended sequence lengths. However, if we prefill the tokens chunk by chunk, we can reduce the $S$ by $C$ times, and thus reduce the activation memory $\mathcal{M}_{\text{act}}$ by $C$ times. Assuming each chunk contains 4096 visual tokens , then $C = \frac{921600}{4096} = 225$ and we can reduce $\mathcal{M}_{\text{act}}$ from 87.9 GB to 0.4 GB.

**KV Cache Memory Analysis.** When using Qwen2.5-VL-7B, with $|V| = 921600$ visual tokens, $|Q| = 256$ text tokens, $L = 28$ layers, $n_{kv} = 4$ heads, and $d_h = 128$, the total memory required to store the KV cache in half precision is:

$$
\begin{aligned}
\text{Memory} &= 2 \times L \times (|V| + |Q|) \times n_{\text{kv}} \times d_h \times 2 \\
&= 52,862,910,464 \text{ bytes} \\
&\approx \boxed{49.2 \text{ GB}}.
\end{aligned}
\tag{13}
$$

## D   DETAIL OF EVALUATION

In this section, we analyze the memory consumption of our approach, focusing on activation memory and key-value (KV) cache memory.

**Implement Setting**

We present the detail of the hyperparameters during the inference in Table 5.

**Evaluation Result**

We present the detail of the evaluation results on StreamingBench in Table 3. StreamAgent achieves state-of-the-art performance among open-source models with an overall score of 57.02, outperforming the recent online model Dispider-7B by 3.90 points (57.02 vs. 53.12). While proprietary model Gemini 1.5 pro leads with 67.07.

**Ablation Setting**

In the experiment of the ablation in streaming KV-cache, we use FPS=0.5 following the setting in ReKV. Notably, we evaluate the streaming KV-cache in offline setting agling with ReKV, the detail of the hyperparameters are presented in Table 4.

| Hyper-parameter | Value |
|---|---|
| *Visual Encoder* | |
| Frame Sampling Rate | FPS=0.5 |
| Input Resolution | 448*448 |
| Visual Tokens per Image | 128 |
| Patch Size | 14x14 |
| *Large Language Model* | |
| Number of Layers | 28 |
| Hidden Size | 3584 |
| Vocabulary Size | 152064 |
| Number of Attention Heads | 28 |
| Number of KV Heads | 4 |
| *Streaming KV-cache* | |
| $\alpha$ | 3 |
| Maximum number of retrievals | 256 |
| GPU-Memory(A100 SXM4) | 80GB |
| *ReKV* | |
| Top-k | 64 |
| Block Size | 1 |
| GPU-Memory(A100 SXM4) | 80GB |

Table 5: The implementation settings of the streaming KV-cache ablation study.

---

**Algorithm 1** Streaming VideoQA with Proactive Response

---

**Require:** Video stream $\mathcal{V}^T = \{v_t\}_{t=1}^T$, question $Q_t$ at time $t$, decision function $D$, response function $f$

**Ensure:** Answer to the question $Q_t$ based on updated memory

1: Initialize memory $m_0 \leftarrow \emptyset$
2: **for** each timestamp $t = 1, 2, \ldots, T$ **do**
3:      Update memory: $m_t \sim p(m_t|m_{t-1}, v_t)$      ▷ Memory update with incoming video clip
4:      Predict future relevant content: $p(E_t|m_t)$      ▷ Predict future event sequence
5:      Plan using *Reactive*, *Proactive*, and *Speculative* modes: $\mathcal{P} \leftarrow \text{Plan}(E_t, S_t)$
6:      Heuristically select optimal plan: $\hat{P} \leftarrow \arg\max_{P_j \in \mathcal{P}} F(P_j)$    ▷ $F(P_j) = G(P_j) + U(P_j)$
7:      **if** $\hat{P}$ requires additional information **then**
8:          Select tools: $\mathcal{T}'_t = \pi_{\text{tool}}(S_t, \hat{P}, Q_t)$
9:          Apply selected tools to video: $v_{t+1} \leftarrow \pi_{\text{crop}}(S_t, \hat{P}, \mathcal{T}'_t)$
10:         Update perception: $S_{t+1} = A(m_t, \mathcal{R}_t)$      ▷ Update state with tool results
11:      **else**
12:          Generate response: $f(Q_t, m_t)$      ▷ Respond with sufficient information
13:      **end if**
14: **end for**
15: **return** Final Answer after reaching $t = T$ or when sufficient information is available

---

# E   PSEUDOCODE

## E.1   ALGORITHM OF THE STREAMAGENT.

We provide the algorithm of the streamAgent in Algorithm 1

---

**Algorithm 2** Streaming KV-cache Algorithm

---

**Require:** Incoming video stream $\{v_1, v_2, \ldots, v_T\}$, query $Q_t$, threshold margin $\alpha$
**Ensure:** Output answer with retrieved memory-aware KV-cache
1: Initialize long-term memory in CPU $\mathcal{M}_{\text{long}} \leftarrow \emptyset$
2: **for** each video clip $v_i$ **do**
3:      Divide $v_i$ into chunks $\mathbf{C}^{v_i} = \{\mathbf{Z}_j^{v_i}\}_{j=1}^{\mu}$
4:      Initialize short-term KV-cache $H_{\text{chunk}}^{k,v} \leftarrow \emptyset$
5:      **for** each chunk $X$ in $\mathbf{C}^{v_i}$ **do**
6:          Compute current $K, V$ using Eq. equation 12
7:          Prefill current chunk X
8:          Append $(K, V)$ to $H_{\text{chunk}}^{k,v}$
9:      **end for**
10:      Store $H_{\text{chunk}}^{k,v}$ into long-term memory: $\mathcal{M}_{\text{long}} \leftarrow \mathcal{M}_{\text{long}} \cup H_{\text{chunk}}^{k,v}$
11: **end for**

12: **Selective Recall**
13: Compute average query attention descriptor: $\mathbf{q}_{\text{avg}} = \frac{1}{T_q} \sum_{j=1}^{T_q} \mathbf{q}_j$
14: Initialize retrieved KV-cache $H_{\text{out}}^{k,v} \leftarrow \emptyset$
15: **for** each layer $l$ **do**
16:      Compute frame-wise attention scores $\mathbf{S}_h$ between $\mathbf{q}_{\text{avg}}$ and stored $\mathbf{k}_j$
17:      Select top frames: $\mathcal{J}_h = \{j \mid \max(\mathbf{S}_h) - \mathbf{S}_{h,j} \leq \alpha\}$
18:      Append corresponding $k_j, v_j$ to $H_{\text{out}}^{k,v}$
19: **end for**
20: Compute final $K, V$ with $H_{\text{out}}^{k,v}$ and $X_q$
21: **return** Answer generated via attention over $(K, V)$

---

## E.2 ALGORITHM OF THE STREAMING KV-CACHE.

We present the algorithm of the streaming KV-cache in Algorithm 2

## F DETAILS OF PROMPT IN STREAMAGENT

> *Incremental Memory*
>
> You're a video understanding assistant. Generate a detailed caption for the current video clip based on the captions of the previous video clips.
> Memory:{memory}

---

**_Future Plan_**

You're a video understanding assistant, and the video input is streamed. You cannot see the entire content at once. To determine whether the video has provided enough information to answer the given question, based on the video, we need to conduct a phased analysis of the task. Please generate a plan for understanding the video. For each plan, follow the steps below:

Step 1: Review Current Information (Confirm what is known and missing)
Based on the video content received so far, answer the following:
* What has happened in the video so far? Briefly list the key events in chronological order.
* Are there any clearly missing or not-yet-shown pieces of key information are still needed in order to answer the question? (For example: the event hasn't fully developed, etc.)
Step 2: Predict Upcoming Developments (Focus on potential clues)
Reasonably predict what might happen next in the video. Write step-by-step what you think is likely to occur:
* What scenes, actions, or turning points are most likely to appear next?
You are not required to give the final judgment or answer—the goal is to outline task predictions to help with better decision-making later.
Please begin the task analysis using the following information:
Question: {question}
Video Memory So Far: {memory}
(The three types reactive, predictive, proactive will be slightly adjusted.)

---

*Heuristic Evaluation*

You are a video understanding assistant. I have provided you with a Question and the Video Memory So Far, followed by two distinct video understanding plans (Plan 1 and Plan 2), each containing a "Step 1: Review Current Information" and a "Step 2: Predict Upcoming Developments." Your task is to evaluate these two plans based on the following criteria:

Accuracy of Current State Analysis (Step 1): How accurately and comprehensively does "Step 1: Review Current Information" (Key Events So Far and Missing Information) reflect the provided Video Memory So Far? Are the identified missing pieces of information truly relevant to answering the Question?

Reasonableness of Future Planning (Step 2): Are the "Step 2: Predict Upcoming Developments" (likely next steps/development sets) logical, plausible, and directly related to the current video memory and the given Question? Do the predictions offer valuable potential clues that could help in answering the Question later?

Finally, compare the two plans and state which one is more reasonable, providing a brief justification for your choice. Please provide your evaluation using the following format:

Question: {question}
Video Memory So Far: {memory}
Plan_list: {plan_list}
Evaluation of Plan 1
Current State Analysis (Step 1) Score: [X/5]
Reasoning: [Your reasoning for the score, commenting on accuracy, completeness, and relevance of missing info.]
Future Planning (Step 2) Score: [Y/5]
Reasoning: [Your reasoning for the score, commenting on logic, plausibility, relevance of clues, and detail.]
Total Score for Plan 1: [X+Y/10]
Evaluation of Plan 2
Current State Analysis (Step 1) Score: [A/5]
Reasoning: [Your reasoning for the score, commenting on accuracy, completeness, and relevance of missing info.]
Future Planning (Step 2) Score: [B/5]
Reasoning: [Your reasoning for the score, commenting on logic, plausibility, relevance of clues, and detail.]
Total Score for Plan 2: [A+B/10]
Conclusion: More Reasonable Plan
[Plan 1 or Plan 2] is more reasonable because [brief justification comparing it to the other plan, highlighting its strengths].

*Trigger*

You are a video understanding assistant receiving streamed video input, meaning you cannot see the entire content at once. Your task is to assess whether the current information is sufficient to answer the given question, or if it is necessary to continue watching the video to obtain missing key details.

Please make your judgment based on the following context:
Question: {question}
Plan_list: {plan}
Video Memory So Far: {memory}
You need to think step-by-step and try your best to give the solution. You should end your solution with 'So the final answer is yes or no'.

*Action Plan*

To assist you in effectively observing and understanding subsequent event changes based on the provided information, I have equipped you with the following tools:
Below are tool descriptions, notes on using tools, and the call command format:
1. No Tool
− Function: Normally accept subsequent video clips without performing any operations.
− Usage: Just specify the tool_name No Tool
− Return Values: Just accept subsequent video clips without performing any operations
2. Zoom In Tool
− Function: Crop and Zoom in subsequent video clip
− Usage: Just specify bbox and the tool_name Zoom In to focus on important area
− Return Values: Area of the video needed to zoom in, give me the bbox coordinates
3. Object Traction Tool
− Function: Object tracing in subsequent video clip
− Usage: Just specify the tool_name Object Tracing to count the object
− Return Values: A list of dictionaries '[xmin, ymin, xmax, ymax]'
4. Detailed Caption Tool
− Function: Select subsequent video clip area and caption
− Usage: Just specify the tool_name Detailed Caption
− Return Values: A list of dictionaries '[xmin, ymin, xmax, ymax]'
5. Object Traction Tool
− Function: Object tracing in subsequent video clip
− Usage: Just specify the tool_name Object Tractuib to count the object
− Return Values: A list of dictionaries '[xmin, ymin, xmax, ymax]'
Example:
The call command for the Crop and Zoom In Tool is:
{{'tool_name': 'Zoom In'}}.
The call command for the Object Tracing Tool is:
{{'tool_name': 'Object Traction', 'object bbox':'{{'object1':[x1min, y1min, x1max, y1max], 'object2:'[x2min, y2min, x2max, y2max], ...}}'}}.
Task Instructions
Please think the following information step by step, formulating an information observation and retrieval plan. Your plan should consider the relationship between the required information, the current video memory, your existing plan list, and the functions of the available tools, with the ultimate goal of effectively observe and understand subsequent event changes and focus on key areas.
After your analysis, output the single best next action in JSON format: '{{'Action': tool_call_command}}'
Current Context * Question: {question} * Plan_list: {plan} * Video Memory So Far: {memory}

## G    LIMITAION

Although our proposed framework and dataset significantly enhance the streaming capabilities of existing offline video large language models (Video-LLMs), there are still some noteworthy limitations.

First, although the decision model is compact and decoupled from the main language model, its decision quality is still constrained by the capacity of the small activation model. While tool usage helps mitigate this issue, the model may still struggle to accurately predict the optimal response timing in highly complex or ambiguous scenarios.

Second, while StreamAgent improves response accuracy through collaborative interactions among multiple agents and proactive acquisition of future information, the lack of large-scale streaming data for training remains a challenge that limits its full potential.

Third, although our streaming KV-cache effectively handles long-term memory, we aim to further reduce computational overhead and improve efficiency by removing redundant information before it enters the MLLM, through various redundancy elimination techniques.

## H  FUTURE WORK

In the future, we aim to use this agent-based workflow to construct data, and then fine-tune models using the constructed data to enhance their tool-using capabilities and their ability to plan for future events. We also intend to apply this approach in real-world scenarios. This planning-then-tool-invocation framework can naturally transfer to practical applications, for example, in intelligent surveillance systems, cameras can proactively adjust their monitoring areas or focal lengths based on the agent's planning to focus on specific regions. Additionally, we hope to leverage more tools specifically designed for video understanding. For instance, when counting repetitions, MLLMs often tend to overcount or undercount. We can use optical flow or other techniques to assist large models in achieving more accurate counting.

## I  ETHICS STATEMENT

This work focuses on developing a model for stream video understanding. The research does not involve human or animal subjects, sensitive personal data, or potentially harmful applications. All datasets used are publicly available benchmark datasets with appropriate licenses. During manuscript preparation, LLM is used solely for grammar correction and language polishing. It did not contribute to the research design, experiments, or conclusions. We are committed to ensuring that our research complies with ethical standards and promotes fairness, transparency, and responsible AI development.

## J  REPRODUCIBILITY

We detail the model design in the paper, including the overview of the StreamAgent framework in Section 2.1 and the streaming video KV cache memory mechanism in Section 2.2. The experimental setup is explained in Section 3.1 and Appendix D. And we provide the details of the prompts used in our experiments in Appendix F. The code is available in the supplementary materials.

