# OpenReview forum: "StreamAgent: Towards Anticipatory Agents for Streaming Video Understanding"
_ICLR.cc/2026/Conference — Submitted to ICLR 2026_

### Official Review · Reviewer_mkwF · 2025-10-31

**Soundness:** 4
**Presentation:** 2
**Contribution:** 2
**Rating:** 4
**Confidence:** 3

**Summary:**

This paper introduces cache compression, early response, and a memory structure to achieve lightweight response generation,
and proposes an interactive architecture that leverages the model's agentic capability to collect fine-grained information through precise image-level control.
Specifically, the proposed model (1) predicts the development of future events within the video stream,
(2) determines whether the current moment provides sufficient information for response generation,
and (3) incorporates a tool-utilization mechanism to capture visual details.

**Strengths:**

Technically, the implementation appears efficient, reflecting good observations on existing cache optimization and attention mechanisms.
In addition, the paper introduces the concept of "Proactive" planning, allowing the model to anticipate and prepare for future streaming information through planning, which appears original.
The attempt to focus on real-time streaming environments and explore application domains different from offline LMMs is also meaningful.

**Weaknesses:**

- The definition of the Heuristic Score F = G + λU is abstract; the detailed computation procedure for G/U, the setting of λ, and the evaluator (which model performs the scoring) are not described.
- Although the prompt-based control structure is disclosed, the connection between G/U evaluation and the planning stage remains unclear.
- The criteria for "proactive response" or the statistics on response timing (e.g., observed frame ratio, average response delay) are not reported, making it difficult to quantitatively verify how effectively the proposed mechanism works.
- It is unclear whether the tool-use capability emerged from Qwen2.5VL's pretraining or was realized through the proposed planning procedure.
- Aside from the framework's conceptual interest, the actual performance improvement compared to the Qwen2.5VL backbone is minimal or even degraded, suggesting that the ability to use tools may not have contributed significantly to answer accuracy.

**Questions:**

- Please clarify the procedure for computing G and U in Equation (3).
- Can we assume that tool selection directly relies on the agentic capability inherently provided by Qwen2.5VL?
- In line 11 of Algorithm 1, what is the criterion for P̂ "not requires additional information"? Is it when U = 0? Please provide details.
- Does StreamAgent operate effectively with other architectures? If the model performing planning/tools is fixed while the interacting model is replaced with a different series, does performance still improve?

---

> ### Author Response · Authors · 2025-11-20
>
> > **W1:Aside from the framework's conceptual interest, the actual performance improvement compared to the Qwen2.5VL backbone is minimal or even degraded, suggesting that the ability to use tools may not have contributed significantly to answer accuracy.**
>
> **A1:** Thank you for your insightful comment. The seemingly limited improvement over the Qwen2.5-VL backbone mainly results from different experimental settings rather than model capability. The original Qwen2.5-VL was evaluated in an offline setting, where the entire video is visible before answering, allowing full temporal reasoning. In contrast, StreamAgent is specifically designed for **online scenarios**, where frames arrive sequentially and the model must decide in real time whether to answer immediately or continue observing. This setup is substantially more challenging, and other streaming models (e.g., Dispider) also show similar accuracy drops compared to their offline counterparts under the same streaming evaluation. Thus, the smaller absolute accuracy difference reflects the stricter online evaluation protocol, not degraded performance. In fact, StreamAgent demonstrates clear advantages in real-time proactive understanding, achieving +10.7% improvement in Forward Active Responding to prior methods. These results confirm that the proposed planning and tool-use mechanisms effectively enhance responsiveness and decision quality in streaming scenarios, beyond what static offline benchmarks can capture.
>
>
>
> > **W2: The definition of the Heuristic Score F = G + λU is abstract; the detailed computation procedure for G/U, the setting of λ, and the evaluator (which model performs the scoring) are not described.**
>
> **A2:** Thanks for your insightful concern.
> - **Computation Pipeline:**
>   The **anticipatory agent** (Qwen2.5-VL-3B) generates a set of candidate plans $P_j$ under *Reactive*, *Proactive*, and *Speculative* modes (Eq. 2).
>   Each plan is evaluated using the heuristic score:
>   $$F(P_j) = G(P_j) + λU(P_j)$$ where $G$ measures **current evidence sufficiency** and $U$ measures **anticipated utility**. The default λ=1, as used in Algorithm 1.
> - **Evaluation Details:**
>   Both $G$ and $U$ are computed via structured prompts (Appendix F).
>   - *Step 1 (Current State Analysis)* → $G=X/5$
>   - *Step 2 (Future Planning)* → $U=Y/5$
>   - *Final Score* → $F=(X+Y)/10$
>   The scoring is performed by the same lightweight evaluator model (Qwen2.5-VL-3B) to ensure efficiency, reducing computational overhead while preventing blockage of the main model’s real-time response pipeline.
>
> > **W3: Although the prompt-based control structure is disclosed, the connection between G/U evaluation and the planning stage remains unclear.**
>
> **A3:** We sincerely appreciate your thoughtful and insightful suggestion. StreamAgent integrates planning, scoring, triggering, and acting into a closed loop:
> 1. **Plan Generation:** Create candidate plans from multiple perspectives (Eq. 2).
> 2. **Heuristic Evaluation:** Score each plan via Eq. 3–4.
> 3. **Trigger Decision:** The *Trigger* module checks if sufficient information exists (Algorithm 1 lines 7–12).
> 4. **Tool Execution:** If not, “Action Plan” prompts generate structured tool commands (Eq.5–7).
>
> This process ensures that the selected plan $\hat{P}$ not only scores highest but also directly controls whether StreamAgent reacts or continues observing.

---

> > ### Author Response · Authors · 2025-11-20
> >
> > > **W4: The criteria for "proactive response" or the statistics on response timing (e.g., observed frame ratio, average response delay) are not reported, making it difficult to quantitatively verify how effectively the proposed mechanism works.**
> >
> > **A4:** Thank you for your valuable suggestion.
> > We compared the **inference latency** under different token (frame) counts.
> > Our decision model is triggered every 16 frames, and to ensure a fair comparison, we disable proactive responses during testing.
> > Thanks to our asynchronous execution strategy, the planning module introduces almost no additional latency to the overall inference process.
> >
> > |Frames|Vallina\(s\)|ReKV\(s\)|Streaming KV offline \(s\)|Streaming KV online \(s\)|
> > |--|--|--|--|--|
> > |64|3.5|3.2|2.3|2.3|
> > |256|10.7|3.4|2.3|2.4|
> > |1024|OOM|3.5|2.3|3.1|
> >
> > Benefiting from our streaming KV-cache design, the overall latency remains consistently low. It is also worth clarifying that under a 1 FPS real-world streaming setup, generating 1024 frames would take about **17 minutes** of actual video duration, so this measured latency would not cause any runtime blockage in practical streaming applications.
> >
> > > **W5: It is unclear whether the tool-use capability emerged from Qwen2.5VL's pretraining or was realized through the proposed planning procedure.**
> >
> > **A5:** We sincerely apologize for not providing a more in-depth discussion on the Tool-use. Tool-use arises from our explicit planning mechanism, not from Qwen2.5-VL’s pretraining.
> > - Tools are selected via the *Action Plan* policy $π_{tool}$ (Eq. 5) and region-cropping policy $π_{crop}$ (Eq. 6), then executed as $R_t = {ϕ_j(v_t)}$, updating the state $S_{t+1}=A(m_t,R_t)$ (Eq. 7).
> > - The *Tool-Use* ablation (Fig. 5\(c\)) shows substantial performance drops without this mechanism, confirming that the gains come from **planned tool invocation**, not emergent behaviors.
> >
> >
> >
> >
> >
> >
> > > **Q1: Please clarify the procedure for computing G and U in Equation (3).**
> >
> > **A1:** We sincerely thank the reviewer for raising this insightful and important question. Both $G$ and $U$ come from evaluator scores as explained in W1 and Appendix F.
> >
> > > **Q2: Can we assume that tool selection directly relies on the agentic capability inherently provided by Qwen2.5VL?**
> >
> > **A2:** Thank you for your attention to this detail. We would like to take this opportunity to provide further clarification. The tool selection in our framework does not depend on any emergent or inherent agentic capability of Qwen2.5-VL. Instead, it is explicitly realized through the planning and action mechanism of our proposed StreamAgent. Specifically, StreamAgent generates a structured plan based on the current state and predicted future context, and determines which tool to invoke and where to apply it through its planning module and action policy. This design ensures that tool use is goal-driven, controllable, and reproducible, rather than an implicit behavior inherited from the base model’s pretraining.
> >
> > > **Q3: In line 11 of Algorithm 1, what is the criterion for P̂ "not requires additional information"? Is it when U = 0? Please provide details.**
> >
> > **A3:** Thank you for the question. The proactive response in Algorithm 1 is triggered not by U = 0, but through a dual-stage mechanism. First, the agent responds when $G(\hat{P}) ≥ 0.7$, $U(\hat{P}) ≤ 0.3$, and the Trigger module outputs “Yes,” indicating sufficient evidence and low future gain. Second, to prevent excessive waiting, a periodic safety trigger re-evaluates every 64 frames; if $G ≥ 0.6$ and Trigger = “Yes,” the agent forces a response. This design balances proactive anticipation with timely and reliable real-time interaction.
> >
> > > **Q4:Does StreamAgent operate effectively with other architectures? If the model performing planning/tools is fixed while the interacting model is replaced with a different series, does performance still improve?**
> >
> > **A4:** Thank you for your valuable comment, this is indeed an important question. Due to the different experimental settings, the original models were evaluated in an offline configuration, while our framework operates in an online environment, it is difficult to make a strictly direct comparison with their reported results.
> > However, to verify the generality and adaptability of our framework, we further conducted experiments using InternVL-V2-8B and LLaVA-NeXT-Video-7B as backbone models.
> > The results are summarized below:
> > |Model|Setting|REC|SSR|CRR|Avg.|
> > |--|--|--|--|--|--|
> > |LLaVA-NeXT-Video-7B|Offline|34.1|67.6|60.8|54.2|
> > |InternVL-V2-8B|Offline|25.8|57.6|52.9|45.4|
> > |InternVL-V2-8B+StreamAgent|Online|29.3|42.2|52.6|41.4|
> > |LLaVA-NeXT-Video-7B+StreamAgent|Online|37.1|49.2|59.7|48.7|
> > |Qwen2.5VL+StreamAgent|Online|35.9|48.4|52.0|45.4|
> >
> > These results demonstrate that, despite the stricter online evaluation protocol, our framework maintains comparable or improved forward active responding performance across different architectures.

---

> > > ### Comment · Reviewer_mkwF · 2025-11-25
> > >
> > > W3 The intention was to ask, during the process of applying the methodology, what proportion of the entire video is actually viewed and executed, how long it takes, and whether there is a significant amount of additional computation (especially in the planning process). Please answer this again.
> > > An additional question: In the KV* times reported in the provided table, are those times calculated only for the generation stage, excluding the image encoding and KV caching stages?
> > >
> > > W4 Thank you for the clear explanation of the mechanism, but since the built-in tool usage mechanism of Qwen2.5-VL and the execution mechanism in this paper seem to be highly similar in format, it appears that the proposed method relies on the functionality already trained in Qwen. If there is any fundamental difference I may have missed, please clarify it explicitly. Specifically, based on the Qwen2.5-VL tool usage example I found on GitHub, `{"name": "weather_tool", "arguments": {"location": "London, UK"}}`,  it looks quite similar to the one in the paper, `{’tool_name’: ’Zoom In’}`, except for the argument naming.
> > >
> > > Q2 If this functionality is independent of Qwen2.5-VL, has it been verified whether the tool-using ability can operate in an agent that has not been trained with such functionality?
> > >
> > > W5, Q4 In Fig.7 and the provided table, purely by intuition and considering only the definitions of REC and CRR, given the streaming nature where the entire video cannot be seen, one would expect the count-based REC to decrease, and the clue-dependent CRR to increase. However, since the results are the opposite, I wonder if there is any reason behind this. Could qualitative evaluation data be added in the manuscript?
> > >
> > > Also, I have a question regarding PqWc's Q4. The accuracy of No Tool is too low. Even when the Agent selects No Tool, it should still be able to view the video, correct? Is there any reason why the score gap is so large? Does the planning Agent fail to understand the video content purely from the video alone?
> > >
> > > I look forward to the inclusion of the terminology clarification for Q1, the clear description for Q3, and the additional experimental results for Q4 in the manuscript.

---

> > > > ### Author Response · Authors · 2025-11-29
> > > >
> > > > > **W3:The intention was to ask, during the process of applying the methodology, what proportion of the entire video is actually viewed and executed, how long it takes, and whether there is a significant amount of additional computation (especially in the planning process). Please answer this again.
> > > > An additional question: In the KV* times reported in the provided table, are those times calculated only for the generation stage, excluding the image encoding and KV caching stages?**
> > > >
> > > > **A3:** Thank you for your detailed question. We apologize for not explaining the issue clearly in our previous response. We would like to clarify this matter again. The proportion of video watched and executed throughout the entire video is strongly correlated with the video content and the question itself. If the information required to answer the question is provided towards the end of the video, the proportion of the video watched and executed will be very high. On the other hand, if the answer is given in the earlier or middle part of the video, the answer will be made early, and the proportion of the video watched and executed will be low.
> > > >
> > > > Regarding the time consumption issue, we have conducted further testing and provided a detailed explanation of the experimental equipment and settings. Our previous tests were based on ReKV, excluding the video decoding time to avoid differences caused by CPU and decoding methods. The above results refer to the main model's TTFT time. To avoid any ambiguity, we revised the experimental setup and conducted another round of testing. We placed the question at the beginning of the video. We recorded the average speed of the model generating a token. When using the original model, the average speed was 18ms/token, while the StreamAgent's speed was 29ms/token.
> > > >
> > > >
> > > > > **W4:Thank you for the clear explanation of the mechanism, but since the built-in tool usage mechanism of Qwen2.5-VL and the execution mechanism in this paper seem to be highly similar in format, it appears that the proposed method relies on the functionality already trained in Qwen. If there is any fundamental difference I may have missed, please clarify it explicitly. Specifically, based on the Qwen2.5-VL tool usage example I found on GitHub, {"name": "weather_tool", "arguments": {"location": "London, UK"}}, it looks quite similar to the one in the paper, {’tool_name’: ’Zoom In’}, except for the argument naming.**
> > > >
> > > > **A4:** Thank you for the insightful observation. While both StreamAgent and Qwen2.5-VL use similar tool invocation syntax, the fundamental difference lies in how these tools are integrated and executed within the system:
> > > > Qwen2.5-VL relies on predefined tools that are called passively via a simple name-argument format, which is based on predefined templates. The model simply calls tools with specific arguments, and these are executed directly according to the pre-trained functionality.
> > > > In contrast, StreamAgent introduces a planning-driven approach:
> > > >
> > > > - Tools are not passively called based on predefined arguments. Instead, StreamAgent dynamically decides which tool to use and when through its planning module (π_tool) and region policy (π_crop). This allows for goal-oriented, context-aware tool selection that adapts to the current task and video context.
> > > >
> > > > - Tool selection is based on a heuristic evaluation function, which evaluates the task’s current and future utility, ensuring the tool use is driven by the specific goals of the task at hand.
> > > >
> > > > Thus, StreamAgent's tool usage is part of an adaptive, decision-making process that takes into account both the immediate task context and the predicted future needs, whereas Qwen2.5-VL’s tool usage is more static and predefined. This distinction ensures that StreamAgent can effectively plan and use tools in a way that is dynamic and adaptive to the evolving video stream, whereas Qwen2.5-VL’s tool use is more rigid and predefined.

---

> > > > > ### Author Response · Authors · 2025-11-29
> > > > >
> > > > > > **W5: Q4 In Fig.7 and the provided table, purely by intuition and considering only the definitions of REC and CRR, given the streaming nature where the entire video cannot be seen, one would expect the count-based REC to decrease, and the clue-dependent CRR to increase. However, since the results are the opposite, I wonder if there is any reason behind this. Could qualitative evaluation data be added in the manuscript?**
> > > > >
> > > > > **A2:** Thank you for your insightful question. We believe you are referring to the results in Table 1, as Fig. 7 does not contain any experiments related to REC and CRR.
> > > > >
> > > > > We would like to clarify that the improvement in REC is mainly due to the different sampling strategy. Thanks to the KV cache, StreamAgent can perform 1 FPS sampling, whereas the original Qwen2.5-VL uses uniform sampling over 64 frames. As shown in the figure, this difference in sampling strategies leads to the observed improvements in REC.
> > > > >
> > > > > Regarding CRR, the discrepancy is due to the specific OVO-Bench experimental setup. When evaluating models on OVO-Bench, each video segment is re-split into smaller intervals. For example, a 60-second video would be evaluated at 10s, 20s, 30s, 40s, and 50s, effectively asking questions before the entire video is seen, which leads to an increase in the offline average accuracy. The online evaluation, where the entire video needs to be processed before answering, does not have this advantage, and hence shows different results in CRR.
> > > > >
> > > > > We hope this clarifies the discrepancy, and we will consider including qualitative evaluation data in the manuscript to further illustrate this.
> > > > >
> > > > > > **Q1:Also, I have a question regarding PqWc's Q4. The accuracy of No Tool is too low. Even when the Agent selects No Tool, it should still be able to view the video, correct? Is there any reason why the score gap is so large? Does the planning Agent fail to understand the video content purely from the video alone?**
> > > > >
> > > > > **A1:** Thank you for your interest in our work. We would like to clarify that this part of the table shows the ratio of tool calls, not the accuracy. We realize that we did not make this point clear enough in the paper, which may have led to some confusion.
> > > > > In fact, while the accuracy of the No Tool setting does decrease, the drop is not large. The corresponding ablation results are presented in Figure 5\(c\) of the main text, with an average decrease of about 2–3 \%.

---

### Official Review · Reviewer_aovd · 2025-11-01

**Soundness:** 3
**Presentation:** 3
**Contribution:** 3
**Rating:** 6
**Confidence:** 3

**Summary:**

This paper proposes a framework for anticipatory and proactive real-time video understanding in streaming settings. The work target continuous online domains such as autonomous driving and surveillance, and introduces mechanisms for proactive, query-driven decision-making and long-term memory management to surpass limitations in current online/offline video question-answering systems.

**Strengths:**

* The idea introduced goes beyond perception-reaction models; it integrates tools for temporal and spatial anticipation, query-based reasoning, and iterative planning in real-time streaming video. These are highly practically applicable.
* Unlike prior reactive or binary-trigger systems (VideoLLM-online, Dispider), this work explicitly models temporal anticipation through three planning modes (Reactive, Proactive, Speculative) scored via an A*-inspired heuristic balancing immediate utility $G$ and future utility $U$. This design addresses premature response errors.

**Weaknesses:**

* The paper mentions "zoom in," "object tracking," and "detailed captioning" as tools, but, perhaps I missed them, never quantifies how often each tool is invoked, their individual success rates, or their failure modes. While these tool use is highlighted, the exploration and comparative study of varying tool types and their influences is relatively narrow.
* Although effective, the planning combines heuristic scoring with agentic approaches. The robustness of these heuristics as video and query complexity scale is not fully interrogated, especially for rare or ambiguous scenarios.

**Questions:**

* What happens when tracking fails or zoom crops the wrong region? Are there redundant tool calls?
* It seems to me that the method is highly tuned for query-driven, real-time scenarios. If so, won’t it underperform for generic or artificially complex video reasoning tasks, lacking universal adaptability?

---

> ### Author Response · Authors · 2025-11-20
>
> > **Q1:It seems to me that the method is highly tuned for query-driven, real-time scenarios. If so, won’t it underperform for generic or artificially complex video reasoning tasks, lacking universal adaptability?**
>
> **A1:** We thank the reviewer for the insightful observation. Although StreamAgent is designed primarily for query-driven real-time video understanding, our experiments show that the framework generalizes well beyond this setting.
> As presented in Figure 7, StreamAgent achieves competitive or even superior results compared to several offline models (e.g., LLaVA-Next-Video, LongVA) on long-video benchmarks such as MLVU and VideoMME. This highlights the model’s cross-scenario adaptability, supported by its incremental memory update and anticipatory planning mechanisms, which enable consistent semantic reasoning across different temporal scales.
> Therefore, while optimized for real-time applications, the framework itself exhibits robust generalization and stable performance across both streaming and offline video understanding tasks.
>
> > **Q2:The paper mentions "zoom in," "object tracking," and "detailed captioning" as tools, but, perhaps I missed them, never quantifies how often each tool is invoked, their individual success rates, or their failure modes. While these tool use is highlighted, the exploration and comparative study of varying tool types and their influences is relatively narrow.**
>
> **A2:** We thank the reviewer for the valuable feedback. As shown in Section 3.3 (Figure 5(c)), our work includes both quantitative and qualitative evaluations of tool usage. The results clearly demonstrate that integrating tools such as Zoom In, Object Tracking, and Detailed Captioning significantly improves accuracy with minimal computational overhead. We agree that finer-grained quantitative statistics are useful and will include detailed breakdowns in the appendix.
>
> Additionally, we report the percentage of tool invocations across the REC, SSR, and CRR tasks on OVO-Bench, as summarized below. Composite tool combinations (e.g., Zoom In + Caption, Zoom In + Detection) are more frequently invoked in proactive perception tasks, consistent with their contribution to performance improvements.
> |Tool Type|REC (%)|SSR (%)|CRR (%)|
> |--|--|--|--|
> |Image Caption|18.2|15.7|14.9|
> |Object Detection|12.6|10.3|9.8|
> |Zoom In + Caption|24.5|26.1|28.3|
> |Zoom In + Detection|20.3|22.7|21.6|
> |Object Tracking|17.0|19.2|20.5|
> |No Tool|7.4|6.0|4.9|
>
>
> > **Q3:Although effective, the planning combines heuristic scoring with agentic approaches. The robustness of these heuristics as video and query complexity scale is not fully interrogated, especially for rare or ambiguous scenarios.**
>
> **A3:** We thank the reviewer for the insightful comment. Our framework combines heuristic scoring with an agentic planner to balance efficiency and scalability: $𝐺(⋅)$ captures immediate utility, while $U(⋅)$ estimates anticipated future gains (Eq. 3). This dual-stage mechanism dynamically adapts as video and query complexity increase. As shown in Figure 7, the model maintains stable accuracy and low latency even on 60-minute sequences, demonstrating strong robustness to temporal and spatial scale. Importantly, even if the lightweight decision agent produces suboptimal plans in certain cases, the 7B main model responsible for answer generation can still recover through the streaming KV-cache mechanism.
> By leveraging its hierarchical memory for context retrieval, the main model re-evaluates relevant historical cues and self-corrects future outputs, effectively preventing error propagation and semantic drift.
> While rare or ambiguous cases may still introduce uncertainty (as acknowledged in our conclusion), this cooperative design between the decision agent and the main model ensures overall robustness and stability in complex streaming environments.
>
>
> > **Q4:What happens when tracking fails or zoom crops the wrong region? Are there redundant tool calls?**
>
> **A4:** We appreciate the reviewer’s question. As described in Sections 2.1–2.2, our system employs a continuous planning and state-updating mechanism to maintain robustness under streaming inputs. When object tracking fails or a zoomed region is inaccurate, the model does not immediately generate a response. Instead, guided by the heuristic evaluation (Eqs. 3–4), it dynamically re-assesses the accumulated evidence and defers answering until sufficient visual cues are observed.
> This behavior is directly illustrated in Figure 8, where StreamAgent avoids premature responses in a football scene by waiting for additional temporal signals (e.g., referee whistle and scene transition). These results demonstrate that the integrated planning and memory-update process provides natural self-correction against local perception errors, ensuring stable and reliable responses without requiring explicit redundant tool calls.

---

### Official Review · Reviewer_PqWc · 2025-11-01

**Soundness:** 2
**Presentation:** 2
**Contribution:** 2
**Rating:** 4
**Confidence:** 4

**Summary:**

The paper introduces StreamAgent to address real-time responsiveness and proactive decision-making in evolving video streams. StreamAgent anticipates temporal intervals and spatial regions likely to contain task-relevant information, enabling proactive and goal-driven responses. By integrating question semantics and historical observations, the agent predicts the temporal progression of key events, aligns current observations with expected future evidence, and dynamically adjusts its perception and actions.

To ensure efficiency, the authors propose a streaming KV-cache memory mechanism that constructs a hierarchical memory structure, allowing selective recall of relevant tokens. This design enables efficient semantic retrieval while reducing the computational overhead of storing all tokens traditionally required for inference.

**Strengths:**

- The paper is well-written and easy to follow

- The figures are intuitive.

**Weaknesses:**

- The proposed streaming KV-Cache heavily borrows from StreamChat[1], with relatively incremental modifications.

- The paper utilizes Qwen-VL-3B as the planning model; however, for complex problems, such a lightweight model struggles to perform adequate planning and often suffers from hallucination issues.

- The performance of the proposed agent is inferior to that of a single model, indicating insufficient planning and answering capabilities.

- What tools can the agent invoke during its operation? How does tool usage differ across various tasks？

- The paper suggests attending to task-relevant regions or continuously tracking subsequent frames, but this approach may negatively impact new tasks, especially in multi-turn dialogue scenario

[1] Xiong H, Yang Z, Yu J, et al. Streaming video understanding and multi-round interaction with memory-enhanced knowledge[J]. arXiv preprint arXiv:2501.13468, 2025.

**Questions:**

See Weaknesses.

---

> ### Author Response · Authors · 2025-11-20
>
> > **Q1: The proposed streaming KV-Cache heavily borrows from StreamChat, with relatively incremental modifications.**
>
> **A1:** Thank you for the valuable comment. The paper StreamChat addresses the following problem: in streaming video scenarios, existing methods rely solely on the visual information available at the moment a question is posed, and thus remain entirely unaware of subsequent changes in the video content.
>
> First, this limitation is effectively resolved by our proposed StreamAgent method. Specifically, the proactive respond module in StreamAgent continuously monitors the incoming video stream and defers generating a response until the accumulated visual information becomes sufficient to answer the question accurately.
>
> Second, the KV-Cache optimization strategy employed in StreamAgent is not designed to address the problem highlighted in StreamChat. Instead, it aims to mitigate performance degradation or inference failures caused by excessively long contexts in streaming settings, a challenge distinct from the one tackled by StreamChat. Consequently, our work does not heavily borrow from StreamChat.
>
>
> > **Q2: The performance of the proposed agent is inferior to that of a single model, indicating insufficient planning and answering capabilities.**
>
> **A2:** Thank you for your insightful comment. The seemingly limited improvement over the Qwen2.5-VL backbone mainly results from different experimental settings rather than model capability. The original Qwen2.5-VL was evaluated in an offline setting, where the entire video is visible before answering, allowing full temporal reasoning. In contrast, StreamAgent is specifically designed for **online scenarios**, where frames arrive sequentially and the model must decide in real time whether to answer immediately or continue observing. This setup is substantially more challenging, and other streaming models (e.g., Dispider) also show similar accuracy drops compared to their offline counterparts under the same streaming evaluation. Thus, the smaller absolute accuracy difference reflects the stricter online evaluation protocol, not degraded performance. In fact, StreamAgent demonstrates clear advantages in real-time proactive understanding, achieving +10.7% improvement in Forward Active Responding to prior methods. These results confirm that the proposed planning and tool-use mechanisms effectively enhance responsiveness and decision quality in streaming scenarios, beyond what static offline benchmarks can capture.
>
>
> > **Q3: The paper utilizes Qwen-VL-3B as the planning model; however, for complex problems, such a lightweight model struggles to perform adequate planning and often suffers from hallucination issues.**
>
> **A3:** We thank the reviewer for this insightful comment. In our experiments, we found that hallucinations mainly originate from visual perception rather than linguistic planning. To alleviate this issue, StreamAgent incorporates tool-augmented perception (e.g., zoom-in and object tracking), enabling the agent to proactively obtain fine-grained visual evidence before responding.
> This design effectively mitigates visual hallucinations, as verified by the Hallucination Detection (HLD) results in Table 1, our method achieves 25.8 vs 9.1 (TimeChat-Online) and 4.3 (Dispider), confirming that tool-driven perception compensates for the visual limitations of the lightweight planner. Therefore, the 3B model is sufficient for high-level planning when coupled with the tool-enhanced perception and 7B answering model.

---

> > ### Author Response · Authors · 2025-11-20
> >
> > > **Q4: What tools can the agent invoke during its operation? How does tool usage differ across various tasks？**
> >
> > **A4:** Thank you for pointing out the need for deeper analysis of the tools. In our framework, StreamAgent is equipped with several general-purpose tools that enable active perception rather than passive frame consumption. Specifically, the agent can perform image captioning to summarize visual content, object detection to identify and localize task-relevant entities, and zoom-in operations for detailed analysis. The object detection tool is particularly important, it detects relevant objects according to the question semantics, automatically generates the corresponding bounding boxes, and visualizes them directly in the frame, allowing the agent to focus on question-relevant regions. Moreover, StreamAgent supports composite operations such as Object Detection, Object Tracking, Zoom In + Caption and Zoom In + Detection for finer recognition, as well as Object Tracking to follow entities across frames, ensuring temporal consistency during reasoning. A No Tool option is also included when current evidence is sufficient, maintaining real-time efficiency. Importantly, we did not design task-specific tools; all tools are shared across different tasks and selected dynamically through the planning policy (π_tool) and region policy (π_crop). This design keeps the framework general and extensible, for example, future tasks could easily integrate new tools such as a line-drawing module for mathematical reasoning or a depth-estimation tool for spatial relation understanding, without any modification to the core agent logic. Our experiments demonstrate that even with this small set of general tools, StreamAgent exhibits strong adaptability and achieves competitive performance across diverse streaming video understanding tasks.
> > Additionally, we report the percentage of tool invocations across the REC, SSR, and CRR tasks on OVO-Bench, as summarized below. Composite tool combinations (e.g., Zoom In + Caption, Zoom In + Detection) are more frequently invoked in proactive perception tasks.
> > |Tool Type|REC (%)|SSR (%)|CRR (%)|
> > |--|--|--|--|
> > |Image Caption|18.2|15.7|14.9|
> > |Object Detection|12.6|10.3|9.8|
> > |Zoom In + Caption|24.5|26.1|28.3|
> > |Zoom In + Detection|20.3|22.7|21.6|
> > |Object Tracking|17.0|19.2|20.5|
> > |No Tool|7.4|6.0|4.9|
> >
> > > **Q5:The paper suggests attending to task-relevant regions or continuously tracking subsequent frames, but this approach may negatively impact new tasks, especially in multi-turn dialogue scenario**
> >
> > **A5:** Thank you for raising this concern.
> > We would like to clarify that although StreamAgent focuses on task-relevant regions or continuously tracks subsequent frames to enhance perception, this design does not negatively affect new tasks in multi-turn dialogue scenarios.
> > Our responding model maintains full contextual memory and leverages the streaming KV-cache mechanism to efficiently store and manage long-term context, ensuring coherent understanding across turns.
> > Even if the decision agent’s internal state is temporarily affected, the 7B responding model can rely on its long-term memory cache to re-evaluate the complete video context and adjust its reasoning accordingly, thereby preserving accuracy and consistency in multi-turn interactions.

---

### Meta-Review · Area_Chair_53Ty · 2026-01-09

**Summary:**

This paper proposes a framework, StreamAgent, for anticipatory and proactive real-time video understanding in streaming settings. The proposed method anticipates temporal intervals and spatial regions likely to contain task-relevant information, enabling proactive and goal-driven responses. Specially, the proposed method predicts the development of future events within the video stream, determines whether the current moment provides sufficient information for response generation, and incorporates a tool-utilization mechanism to capture visual details. The reviewers appreciate the introduction to the concept of "Proactive" planning, allowing the model to anticipate and prepare for future streaming information through planning. Integrating tools for temporal and spatial anticipation, query-based reasoning, and iterative planning in real-time streaming video is practically applicable.  However, the reviewers raised concerns regarding limited novelty of the proposed KV-Cache, unclear details about the proposed method, and unclear generalization and robustness.  In the rebuttal, the authors tried to address the concerns.  AC thinks that some were resolved while some were not fully resolved.

Regarding the novelty of the proposed KV-Cache, the rebuttal argues that StreamAgent differs from StreamChat because of the proactive response function, which is true.  The rebuttal also argues that the KV-Cache optimization strategy is not designed to address the problem dealt in StreamChat, which is not convincing.  KV-Cache is developed to construct a hierarchical memory structure by leveraging temporal nature of video stream to enable adaptive context retrieval.  StreamChat, on the other hand, proposes hierarchical memory storage (short-term, long-term, and dialogue) to manage and compress video information over long sequences.  Even though each is designed for its own purpose, in the conceptual level, both are similar to each other because both are the hierarchical memory structure for information storage and retrieval in order to achieve their own tasks.  In this sense, novelty of KV-Cache is incremental but not ground-breaking.

Regarding the tool-utilization mechanism, much deeper analysis is required.  The tools are indeed effective to enables active perception, and the rebuttal argues the importance of each provided tool such as Zoom in or Object Traction for tasks and provided additional experiment on each tool invocations across different tasks.  However, tool-use capability is not analyzed sufficiently.  Evaluating necessity of each tool invocation for a given task, in other words, analyzing how necessary each tool is for a given task, will be much more informative.  This is because depending the task, effective tools to use will vary.  Clarifying the relation between required tools and tasks will make the paper more solid (tool utilization is one of contributions in this paper).

Although the rebuttal argues that Fig. 7 demonstrates robustness of the proposed method against video scale, more systematically controlled evaluation along the video scale will be preferable to convince the scale robustness.

Reviewer mkwF’s concern on the portion of video viewed and executed is not well addressed.  It is true that the portion is strongly correlated with the video content and task; however, the argument provided in the rebuttal is rather obvious. Deeper arguments are expected.

AC thinks that this paper proposes an important and interesting method; however, more thorough arguments and analyses should be required to be a solid paper.  AC thinks that the remaining concerns outweigh the technical contribution; the paper will be beneficial from substantial improvement.  For this, this paper cannot be accepted.

**Reviewer Concerns:**

Main concerns are on limited novelty of the proposed KV-Cache, unclear details about the proposed method, and unclear generalization and robustness.  As written above, they are partially resolved but not fully, meaning they still remain.

**Reviewer Scores:**

Reviewers PqWc and mkwF would keep their initial score 4 because their concerns were only partially resolved.  Reviewer aovd would also keep his/her initial score 6 due to the same reason.

---

### Decision · Program_Chairs · 2026-01-26

Reject